# Online Matrix Completion with Side Information

**Mark Herbster, Stephen Pasteris, Lisa Tse**
Department of Computer Science
University College London
London WC1E 6BT, England, UK
{m.herbster,s.pasteris,l.tse}@cs.ucl.ac.uk

## Abstract

We give an online algorithm and prove novel mistake and regret bounds for online binary matrix completion with side information. The mistake bounds we prove are of the form $\tilde{\mathcal{O}}(\frac{\mathcal{D}}{\gamma^2})$. The term $\frac{1}{\gamma^2}$ is analogous to the usual margin term in SVM (perceptron) bounds. More specifically, if we assume that there is some factorization of the underlying $m \times n$ matrix into $\boldsymbol{PQ}^\top$, where the rows of $\boldsymbol{P}$ are interpreted as "classifiers" in $\Re^d$ and the rows of $\boldsymbol{Q}$ as "instances" in $\Re^d$, then $\gamma$ is is the maximum (normalized) margin over all factorizations $\boldsymbol{PQ}^\top$ consistent with the observed matrix. The quasi-dimension term $\mathcal{D}$ measures the quality of side information. In the presence of vacuous side information, $\mathcal{D} = m + n$. However, if the side information is predictive of the underlying factorization of the matrix, then in an ideal case, $\mathcal{D} \in \mathcal{O}(k + \ell)$ where $k$ is the number of distinct row factors and $\ell$ is the number of distinct column factors. We additionally provide a generalization of our algorithm to the inductive setting. In this setting, we provide an example where the side information is not directly specified in advance. For this example, the quasi-dimension $\mathcal{D}$ is now bounded by $\mathcal{O}(k^2 + \ell^2)$.

## 1   Introduction

We consider the problem of online binary matrix completion with *side information*. In our setting the *learner* receives data sequentially, so that on a trial $t = 1, \ldots, T$: 1) the learner is queried by the *environment* to predict matrix entry $(i_t, j_t)$; 2) the learner predicts a label $\hat{y}_t \in \{-1, 1\}$; 3) the learner receives a label $y_t \in \{-1, 1\}$ from the environment and 4) a mistake is incurred if $y_t \neq \hat{y}_t$. There are no probabilistic assumptions on how the environment generates its instances or their labels; it is an arbitrary process which in fact may be adversarial. The only restriction on the environment is that it does not "see" the learner's $\hat{y}_t$ until after it reveals $y_t$. The learner's aim will be to minimize its *expected regret*, $\sum_{t=1}^T \mathbb{E}[y_t \neq \hat{y}_t] - \min_{\boldsymbol{U}} \sum_{t=1}^T [y_t \neq U_{i_t j_t}]$, where the minimization is over $\boldsymbol{U} \in \{-1, 1\}^{m \times n}$ and the expectation is with respect to the learner's internal randomization. We will also consider *mistake bounds* where the aim is to minimize the learner's mistakes under the assumption (*realizability*) that there exists a $\boldsymbol{U}$ such that $U_{i_t j_t} = y_t$ for all $t \in [T]$. To aid the learner, side information is associated with each row and column. For instance, in the classic "Netflix challenge" [1], the rows of the matrix correspond to viewers and the columns to movies, with entries representing movie ratings. It is natural to suppose that we have side information in the form of demographic information for each user, and metadata for the movies. In this work, we consider both *transductive* and *inductive* models. In the former model, the side information associated with each row and column is specified completely in advance in the form of a pair of positive definite matrices that inform similarity between row pairs and column pairs. For the inductive model, a pair of kernel functions is specified over potentially continuous domains, where one is for the rows and the other is for the columns. What is not specified is the mapping from the domain of the kernel function to

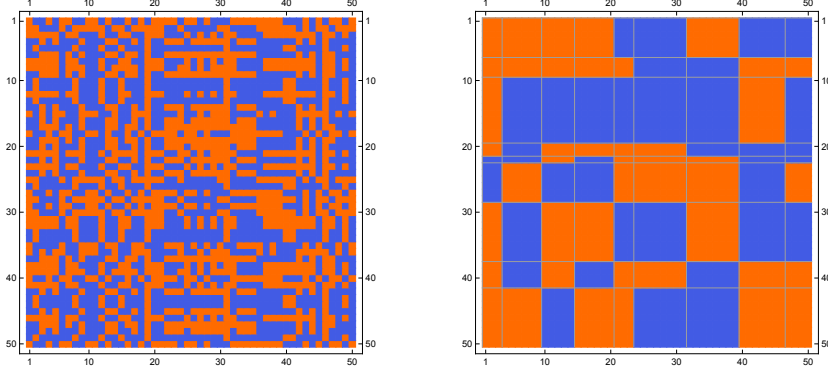

Figure 1: A $(9, 9)$-biclustered $50 \times 50$ binary matrix before/after permuting into latent blocks.

specific rows or columns, which is only revealed sequentially. In the Netflix example, the inductive model is especially natural if new users or movies are introduced during the learning process.

In Theorem 1, we will give regret and mistake bounds for online binary matrix completion with side information. Although this theorem has a broader applicability, our interpretation will focus on the case that the matrix has a *latent block structure*. Hartigan [2] introduced the idea of permuting a matrix by both the rows and columns into a few homogeneous blocks. This is equivalent to assuming that each row (column) has an associated row (column) class, and that the matrix entry is completely determined by its corresponding row and column classes. This has since become known as co- or bi-clustering. This same assumption has become the basis for probabilistic models which can then be used to "complete" a matrix with missing entries. The authors of [3] give rate-optimal results for this problem in the batch setting and provide a literature overview. It is natural to compare this assumption to the dominant alternative, which assumes that there exists a low rank decomposition of the matrix to be completed, see for instance [4]. Common to both approaches is that associated with each row and column, there is an underlying *latent* factor so that the given matrix entry is determined by a function on the appropriate row and column factor. The low-rank assumption is that the latent factors are vectors in $\Re^d$ and that the function is the dot product. The latent block structure assumption is that the latent factors are instead categorical and that the function between factors is arbitrary. In this work we prove mistake bounds of the form $\tilde{\mathcal{O}}(\mathcal{D}/\gamma^2)$. The term $1/\gamma^2$ is a parameter of our algorithm which serves as upper estimate of the squared margin complexity $\mathrm{mc}(\boldsymbol{U})^2$ of the comparator matrix $\boldsymbol{U}$. The notion of margin complexity in machine learning was introduced in [5], where it was used to study the learnability of concept classes via linear embeddings. It was further studied in [6], and in [7] a detailed study of margin complexity, trace complexity and rank in the context of statistical bounds for matrix completion was given. The squared margin complexity is upper bounded by rank. Furthermore, if our $m \times n$ matrix has a latent block structure with $k \times \ell$ homogeneous blocks (for an illustration, see Figure 1) , then $\mathrm{mc}(\boldsymbol{U})^2 \leq \min(k, \ell)$. The second term in our bound is the quasi-dimension $\mathcal{D}$ which, to the best of our knowledge, is novel to this work. The quasi-dimension measures the extent to which the side information is "predictive" of the comparator matrix. In Theorem 3, we provide an upper bound on the quasi-dimension, which measures the predictiveness of the side information when the comparator matrix has a latent block structure. If there is only vacuous side information, then $\mathcal{D} = m + n$. However, if there is a $k \times \ell$ latent block structure and the side information is predictive, then $\mathcal{D} \in \mathcal{O}(k + \ell)$; hence our nomenclature "quasi-dimension." In this case, we then have that the mistake bound term $\mathcal{D}/\gamma^2 \in \mathcal{O}(k\ell)$, which we will later argue is optimal. Although latent block structure may appear to be a "fragile" measure of matrix complexity, our regret bound implies that performance will scale smoothly in the case of adversarial noise.

The paper is organized as follows. First, we discuss related literature. We then introduce preliminary concepts in Section 2. In Section 3, we present our online matrix completion algorithm as well as a theorem (Theorem 1) that characterizes its performance in the transductive setting. In Section 4, we formally introduce the concept of latent block structure (Definition 2) and provide an upper bound (Theorem 3) for the quasi-dimension $\mathcal{D}$ when the matrix has latent block structure. We then provide an example that bounds $\mathcal{D}$ when we have graph-based side information (Section 4.1); and a further example (Section 4.2) when the matrix has additionally a "community" structure. Finally, in Section 5, we present an algorithm for the inductive setting, as well as an example illustrating a

bound on $\mathcal{D}$ when the side information comes as vectors in $\Re^d$ which are separated by a clustering via hyper-rectangles. Proofs as well as an experiment on synthetic data are contained in the appendices.

**Related literature**

Matrix completion has been studied extensively in the batch setting, see for example [8; 9; 10; 11] and references therein. Central to these approaches is the aim of finding a low-rank factorization by optimizing a convex proxy to rank, such as the trace norm [12]. The papers [13; 14; 15; 16] are partially representative of methods to incorporate side-information into the matrix completion task. The inductive setting for matrix completion has been studied in [13] through the use of tensor product kernels, and [17] takes a non-convex optimization approach. Some examples in the transductive setting include [15; 16]. The last two papers use graph Laplacians to model the side information, which is similar to our approach. To achieve this, two graph Laplacians are used to define regularization functionals for both the rows and the columns so that rows (columns) with similar side information tend to have the same values. In particular, [16] resembles our approach by applying the Laplacian regularization functionals to the underlying row and column factors directly. An alternate approach is taken in [15], where the regularization is instead applied to the row space (column space) of the "surface" matrix.

In early work, the authors of [18; 19] proved mistake bounds for learning a binary relation which can be viewed as a special case of matrix completion. In the regret setting, with minimal assumptions on the loss function, the regret of the learner is bounded in terms of the *trace-norm* of the underlying comparator matrix in [20]. The authors of [21] provided tight upper and lower bounds in terms of a parameterized complexity class of matrices that include the bounded-trace-norm and bounded-max-norm matrices as special cases. None of the above references considered the problem of side information. The results in [22; 23; 24] are nearest in flavor to the results given here. In [24], a mistake bound of $\tilde{\mathcal{O}}((m + n)\,\mathrm{mc}(\boldsymbol{U})^2)$ was given. Latent block structure was also introduced to the online setting in [24]; however, it was treated in a limited fashion and without the use of side information. The papers [22; 23] both used side information to predict a limited complexity class of matrices. In [22], side information was used to predict if vertices in a graph are "similar"; in Section 4.2 we show how this result can be obtained as a special case of our more general bound. In [23], a more general setting was considered, which as a special case addressed the problem of a switching graph labeling. The model in [23] is considerably more limited in its scope than our Theorem 1. To obtain our technical results, we used an adaptation of the matrix exponentiated gradient algorithm [25]. The general form of our regret bound comes from a matricization of the regret bound proven for a Winnow-inspired algorithm [26] for linear classification in the vector case given in [27]. For a more detailed discussion, see Appendix B.2.

## 2   Preliminaries

For any positive integer $m$, we define $[m] := \{1, 2, \ldots, m\}$. For any predicate $[\text{PRED}] := 1$ if PRED is true and equals 0 otherwise, and $[x]_+ := x[x > 0]$. We denote the inner product of vectors $\boldsymbol{x}, \boldsymbol{w} \in \Re^n$ as $\langle \boldsymbol{x}, \boldsymbol{w} \rangle = \sum_{i=1}^n x_i w_i$ and the norm as $\|\boldsymbol{w}\| = \sqrt{\langle \boldsymbol{w}, \boldsymbol{w} \rangle}$. The $ith$ coordinate $m$-dimensional vector is denoted $\boldsymbol{e}_m^i := ([j = i])_{j \in [m]}$; we will often abbreviate the notation and use $\boldsymbol{e}^i$ on the assumption that the dimensionality of the space may be inferred. For vectors $\boldsymbol{p} \in \Re^m$ and $\boldsymbol{q} \in \Re^n$ we define $[\boldsymbol{p}; \boldsymbol{q}] \in \Re^{m+n}$ to be the concatenation of $\boldsymbol{p}$ and $\boldsymbol{q}$, which we regard as a column vector. Hence $[\boldsymbol{p}; \boldsymbol{q}]^\top [\bar{\boldsymbol{p}}; \bar{\boldsymbol{q}}] = \boldsymbol{p}^\top \bar{\boldsymbol{p}} + \boldsymbol{q}^\top \bar{\boldsymbol{q}}$. We let $\Re^{m \times n}$ be the set of all $m \times n$ real-valued matrices. If $\boldsymbol{X} \in \Re^{m \times n}$ then $\boldsymbol{X}_i$ denotes the $i$-th $n$-dimensional row vector and the $(i, j)^{th}$ entry of $\boldsymbol{X}$ is $X_{ij}$. We define $\boldsymbol{X}^+$ and $\boldsymbol{X}^\top$ to be its pseudoinverse and transpose, respectively. The trace norm of a matrix $\boldsymbol{X} \in \Re^{m \times n}$ is $\|\boldsymbol{X}\|_1 = \mathrm{tr}(\sqrt{\boldsymbol{X}^\top \boldsymbol{X}})$, where $\sqrt{\cdot}$ indicates the unique positive square root of a positive semi-definite matrix, and $\mathrm{tr}(\cdot)$ denotes the trace of a square matrix. This is given by $\mathrm{tr}(\boldsymbol{Y}) = \sum_{i=1}^n Y_{ii}$ for $\boldsymbol{Y} \in \Re^{n \times n}$. The $m \times m$ identity matrix is denoted $\boldsymbol{I}^m$. In addition, we define $\boldsymbol{S}^m$ to be the set of $m \times m$ symmetric matrices and let $\boldsymbol{S}_+^m$ and $\boldsymbol{S}_{++}^m$ be the subset of positive semidefinite and strictly positive definite matrices respectively. Recall that the set of symmetric matrices $\boldsymbol{S}_+^m$ has the following partial ordering: for every $\boldsymbol{M}, \boldsymbol{N} \in \boldsymbol{S}_+^m$, we say that $\boldsymbol{M} \preceq \boldsymbol{N}$ if and only if $\boldsymbol{N} - \boldsymbol{M} \in \boldsymbol{S}_+^m$. We also define the squared radius of $\boldsymbol{M} \in \boldsymbol{S}_+^m$ as $\mathcal{R}_{\boldsymbol{M}} := \max_{i \in [m]} M_{ii}^+$.

For every matrix $\boldsymbol{U} \in \Re^{m \times n}$, we define $\mathrm{SP}(\boldsymbol{U}) = \{\boldsymbol{V} \in \Re^{m \times n} : \forall_{ij} V_{ij} U_{ij} > 0\}$, the set of matrices which are sign consistent with $\boldsymbol{U}$. We also define $\mathrm{SP}^1(\boldsymbol{U}) = \{\boldsymbol{V} \in \Re^{m \times n} : \forall_{ij} V_{ij}\,\mathrm{sign}(U_{ij}) \geq 1\}$, that is the set of matrices which are sign consistent with $\boldsymbol{U}$ with a margin of at least one. We now

introduce complexity measures for matrices. The max-norm (or $\gamma_2$ norm [6]) of a matrix $\boldsymbol{U} \in \Re^{m \times n}$ is defined by

$$\|\boldsymbol{U}\|_{\max} := \min_{\boldsymbol{P}\boldsymbol{Q}^\top = \boldsymbol{U}} \left\{ \max_{1 \le i \le m} \|\boldsymbol{P}_i\| \times \max_{1 \le j \le n} \|\boldsymbol{Q}_j\| \right\}, \tag{1}$$

where the minimum is over all matrices $\boldsymbol{P} \in \Re^{m \times d}$, $\boldsymbol{Q} \in \Re^{n \times d}$ and every integer $d$. Instead of constraining the dimensionality of $\boldsymbol{P}_i$ and $\boldsymbol{Q}_i$ as is done with rank, it constraints their maximum $\ell_2$-norms. The *margin complexity* of a matrix $\boldsymbol{U} \in \Re^{m \times n}$ is

$$\mathrm{mc}(\boldsymbol{U}) := \min_{\boldsymbol{V} \in \mathrm{SP}^1(\boldsymbol{U})} \|\boldsymbol{V}\|_{\max} = \min_{\boldsymbol{P}\boldsymbol{Q}^\top \in \mathrm{SP}(\boldsymbol{U})} \max_{ij} \frac{\|\boldsymbol{P}_i\| \, \|\boldsymbol{Q}_j\|}{|\langle \boldsymbol{P}_i, \boldsymbol{Q}_j \rangle|}. \tag{2}$$

This is the counterpart of the max-norm to learn a sign matrix. Observe that for $\boldsymbol{U} \in \{-1, 1\}^{m \times n}$, $1 \le \mathrm{mc}(\boldsymbol{U}) \le \|\boldsymbol{U}\|_{\max} \le \min(\sqrt{m}, \sqrt{n})$, where the lower bound follows from the right hand side of (2) and the upper bound follows since we may decompose $\boldsymbol{U} = \boldsymbol{U}\boldsymbol{I}^n$ or as $\boldsymbol{U} = \boldsymbol{I}^m \boldsymbol{U}$. Note there may be a large gap between the margin complexity and the max-norm. In [6] a matrix in $\boldsymbol{U} \in \{-1, 1\}^{n \times n}$ was given such that $\mathrm{mc}(\boldsymbol{U}) = \log n$ and $\|\boldsymbol{U}\|_{\max} = \Theta(\sqrt{n}/\log n)$. We denote the classes of $m \times d$ *row-normalized* and *block expansion* matrices as $\mathcal{N}^{m,d} := \{\hat{\boldsymbol{P}} \subset \Re^{m \times d} : \|\hat{\boldsymbol{P}}_i\| = 1, i \in [m]\}$ and $\mathcal{B}^{m,d} := \{\boldsymbol{R} \subset \{0, 1\}^{m \times d} : \|\boldsymbol{R}_i\| = 1, i \in [m], \mathrm{rank}(\boldsymbol{R}) = d\}$, respectively. Block expansion matrices may be seen as a generalization of permutation matrices, additionally duplicating rows (columns) by left (right) multiplication. We define the *quasi-dimension* of a matrix $\boldsymbol{U} \in \Re^{m \times n}$ with respect to $\boldsymbol{M} \in \boldsymbol{S}_{++}^m$, $\boldsymbol{N} \in \boldsymbol{S}_{++}^n$ at margin $\gamma$ as

$$\mathcal{D}_{\boldsymbol{M}, \boldsymbol{N}}^\gamma(\boldsymbol{U}) := \min_{\hat{\boldsymbol{P}}\hat{\boldsymbol{Q}}^\top = \gamma \boldsymbol{U}} \mathcal{R}_{\boldsymbol{M}} \, \mathrm{tr}\left(\hat{\boldsymbol{P}}^\top \boldsymbol{M} \hat{\boldsymbol{P}}\right) + \mathcal{R}_{\boldsymbol{N}} \, \mathrm{tr}\left(\hat{\boldsymbol{Q}}^\top \boldsymbol{N} \hat{\boldsymbol{Q}}\right), \tag{3}$$

where the infimum is over all row-normalized matrices $\hat{\boldsymbol{P}} \in \mathcal{N}^{m,d}$ and $\hat{\boldsymbol{Q}} \in \mathcal{N}^{n,d}$ and every integer $d$. If the infimum does not exist then $\mathcal{D}_{\boldsymbol{M}, \boldsymbol{N}}^\gamma(\boldsymbol{U}) := +\infty$. Note that the infimum exists iff $\|\boldsymbol{U}\|_{\max} \le 1/\gamma$. The quasi-dimension quantifies how aligned the rows of $\hat{\boldsymbol{P}}$ ($\hat{\boldsymbol{Q}}$) are with the matrices $\boldsymbol{M}$ ($\boldsymbol{N}$). Note that $\mathcal{D}_{\boldsymbol{M}, \boldsymbol{N}}^\gamma(\boldsymbol{U}) = m + n$ if $\|\boldsymbol{U}\|_{\max} \le 1/\gamma$, $\boldsymbol{M} = \boldsymbol{I}^m$ and $\boldsymbol{N} = \boldsymbol{I}^n$.

We now introduce notation specific to the graph setting. Let $\mathcal{G}$ be an $m$-vertex connected, undirected graph with positive weights. The Laplacian $\boldsymbol{L}$ of $\mathcal{G}$ is defined as $\boldsymbol{D} - \boldsymbol{A}$, where $\boldsymbol{D}$ is the $m \times m$ degree matrix and $\boldsymbol{A}$ is the $m \times m$ adjacency matrix. Observe that as $\mathcal{G}$ is connected, $\boldsymbol{L}$ is a rank $m - 1$ matrix with $\boldsymbol{1}$ in its null space. From $\boldsymbol{L}$ we define the (strictly) positive definite *PDLaplacian* $\boldsymbol{L}^\circ := \boldsymbol{L} + \left(\frac{1}{m}\right)\left(\frac{1}{m}\right)^\top \mathcal{R}_{\boldsymbol{L}}^{-1}$. Observe that if $\boldsymbol{u} \in [-1, 1]^m$ then $(\boldsymbol{u}^\top \boldsymbol{L}^\circ \boldsymbol{u})\mathcal{R}_{\boldsymbol{L}^\circ} \le 2(\boldsymbol{u}^\top \boldsymbol{L} \boldsymbol{u} \, \mathcal{R}_{\boldsymbol{L}} + 1)$, and similarly, $(\boldsymbol{u}^\top \boldsymbol{L} \boldsymbol{u})\mathcal{R}_{\boldsymbol{L}} \le \frac{1}{2}(\boldsymbol{u}^\top \boldsymbol{L}^\circ \boldsymbol{u})\mathcal{R}_{\boldsymbol{L}^\circ}$ (see [28] for details of this construction).

## 3 Transductive Matrix Completion

---

**Algorithm 1** Predicting a binary matrix with side information in the transductive setting.

---

**Parameters:** Learning rate: $0 < \eta$, quasi-dimension estimate: $1 \le \widehat{\mathcal{D}}$, margin estimate: $0 < \gamma \le 1$, non-conservative flag [NON-CONSERVATIVE] $\in \{0, 1\}$ and side information matrices $\boldsymbol{M} \in \boldsymbol{S}_{++}^m$, $\boldsymbol{N} \in \boldsymbol{S}_{++}^n$ with $m + n \ge 3$

**Initialization:** $\mathbb{M} \leftarrow \emptyset$ ; $\tilde{\boldsymbol{W}}^1 \leftarrow \frac{\widehat{\mathcal{D}}}{(m+n)} \boldsymbol{I}^{m+n}$.

**For** $t = 1, \ldots, T$

- Receive pair $(i_t, j_t) \in [m] \times [n]$.
- Define
$$\tilde{\boldsymbol{X}}^t := \boldsymbol{x}^t(\boldsymbol{x}^t)^\top := \left[\frac{\sqrt{\boldsymbol{M}^+}\boldsymbol{e}_m^{i_t}}{\sqrt{2\mathcal{R}_{\boldsymbol{M}}}}; \frac{\sqrt{\boldsymbol{N}^+}\boldsymbol{e}_n^{j_t}}{\sqrt{2\mathcal{R}_{\boldsymbol{N}}}}\right]\left[\frac{\sqrt{\boldsymbol{M}^+}\boldsymbol{e}_m^{i_t}}{\sqrt{2\mathcal{R}_{\boldsymbol{M}}}}; \frac{\sqrt{\boldsymbol{N}^+}\boldsymbol{e}_n^{j_t}}{\sqrt{2\mathcal{R}_{\boldsymbol{N}}}}\right]^\top. \tag{4}$$
- Predict
$$Y_t \sim \mathrm{UNIFORM}(-\gamma, \gamma) \times [\text{NON-CONSERVATIVE}] \, ; \; \bar{y}_t \leftarrow \mathrm{tr}\left(\tilde{\boldsymbol{W}}^t \tilde{\boldsymbol{X}}^t\right) - 1 \, ; \; \hat{y}_t \leftarrow \mathrm{sign}(\bar{y}_t - Y_t) \, .$$
- Receive label $y_t \in \{-1, 1\}$.
- If $y_t \ne \hat{y}_t$ then $\mathbb{M} \leftarrow \mathbb{M} \cup \{t\}$.
- If $y_t \bar{y}_t < \gamma \times [\text{NON-CONSERVATIVE}]$ then
$$\tilde{\boldsymbol{W}}^{t+1} \leftarrow \exp\left(\log(\tilde{\boldsymbol{W}}^t) + \eta y_t \tilde{\boldsymbol{X}}^t\right).$$
- Else $\tilde{\boldsymbol{W}}^{t+1} \leftarrow \tilde{\boldsymbol{W}}^t$.

---

Algorithm 1 corresponds to an adapted MATRIX EXPONENTIATED GRADIENT (MEG) algorithm [25] to perform transductive matrix completion with side information. Although the algorithm is a special case of MEG, the following theorem does not follow as a special case of the analysis in [25]. The underlying proof techniques are discussed in further detail in Appendix B.2. The following theorem includes an expected regret bound and a mistake bound, where the regret bound is the more flexible, noise-tolerant bound. Hence, in practice, the algorithm should always be run non-conservatively to obtain the regret bound, as per the theorem assumptions. We only include the conservative case for further insight as the bound is simpler, being analogous to Novikoff's (perceptron) bound. In this case, where we also assume realizability and exact tuning ($\gamma = 1/\operatorname{mc}(\boldsymbol{U})$), the mistakes are bounded by $\tilde{\mathcal{O}}(\mathcal{D}\operatorname{mc}(\boldsymbol{U})^2)$. The term $\mathcal{D}$ evaluates the predictive quality of the side information provided to the algorithm. In order to evaluate $\mathcal{D}$, we provide an upper bound in Theorem 3 that is more straightforward to interpret. Examples are given in Sections 4.1 and 5.1, where Theorem 3 is applied to evaluate the quality of side information in idealized scenarios. The $\operatorname{mc}(\boldsymbol{U})$ term is analogous to the inverse margin in perceptron, SVM, and other "largin margin" classifiers.

**Theorem 1.** *The expected regret of Algorithm 1 with* **non-conservative** *updates (*[NON-CONSERVATIVE] $= 1$*) and parameters* $\gamma \in (0,1]$, $\widehat{\mathcal{D}} \geq \mathcal{D}_{\boldsymbol{M},\boldsymbol{N}}^{\gamma}(\boldsymbol{U})$, $\eta = \sqrt{\frac{\widehat{\mathcal{D}}\log(m+n)}{2T}}$, *p.d. matrices* $\boldsymbol{M} \in \boldsymbol{S}_{++}^m$ *and* $\boldsymbol{N} \in \boldsymbol{S}_{++}^n$ *is bounded by*

$$\mathbb{E}[|\mathbb{M}|] - \sum_{t \in [T]} [y_t \neq U_{i_t j_t}] \leq 4\sqrt{2(\widehat{\mathcal{D}}/\gamma^2)\log(m+n)T} \tag{5}$$

*for all* $\boldsymbol{U} \in \{-1,1\}^{m \times n}$ *with* $\|\boldsymbol{U}\|_{max} \leq 1/\gamma$.

*The mistakes in the* **realizable** *case with* **conservative** *updates (*[NON-CONSERVATIVE] $= 0$*) and parameters* $1/\eta = 1/\gamma \geq \operatorname{mc}(\boldsymbol{U})$, $\widehat{\mathcal{D}} \geq \min_{\boldsymbol{V} \in \operatorname{SP}^1(\boldsymbol{U})} \mathcal{D}_{\boldsymbol{M},\boldsymbol{N}}^{\gamma}(\boldsymbol{V})$ *are bounded by,*

$$|\mathbb{M}| \leq 3.6(\widehat{\mathcal{D}}/\gamma^2)\log(m+n), \tag{6}$$

*for all* $\boldsymbol{U} \in \{-1,1\}^{m \times n}$ *with* $\operatorname{mc}(\boldsymbol{U}) \leq 1/\gamma$ *and* $y_t = U_{i_t j_t}$ *for all* $t \in \mathbb{M}$.

If the side information is vacuous, that is $\boldsymbol{M} = \boldsymbol{I}^m$ and $\boldsymbol{N} = \boldsymbol{I}^n$, then $\mathcal{D} = m + n$. In this scenario, we recover a special case[1] of the analysis of [21] up to constant factors, then with the additional assumption of realizability we recover [24, Theorem 3.1]. The term $\mathcal{D}$ is difficult to directly quantify. In the next section, we specialize our analysis to the case that the matrix $\boldsymbol{U}$ has a latent block structure.

## 4 Latent Block Structure

We introduce the concept class of $(k,\ell)$-binary-biclustered matrices (previously defined in [24, Section 5]), in the following definition. We then give an upper bound to $\mathcal{D}_{\boldsymbol{M},\boldsymbol{N}}^{\gamma}(\boldsymbol{U})$ when a matrix has this type of latent structure in Theorem 3. The magnitude of the bound will depend on how "predictive" matrices $\boldsymbol{M}$ and $\boldsymbol{N}$ are of the latent block structure. In Sections 4.1 and 4.2, we will use a variant of the discrete Laplacian matrix for $\boldsymbol{M}$ and $\boldsymbol{N}$ to encode side information and illustrate the resultant bounds for idealized scenarios.

**Definition 2.** *The class of* $(k,\ell)$-*binary-biclustered matrices is defined as*

$$\mathbb{B}_{k,\ell}^{m,n} = \{\boldsymbol{U} \in \{-1,1\}^{m \times n} : \boldsymbol{r} \in [k]^m, \boldsymbol{c} \in [\ell]^n, \boldsymbol{U}^* \in \{-1,1\}^{k \times \ell}, U_{ij} = U_{r_i c_j}^*, i \in [m], j \in [n]\}.$$

Thus each row $r_i$ is associated with a latent factor in $[k]$ and each column $c_j$ is associated with a latent factor in $[\ell]$ and the interaction of factors is determined by a matrix $\boldsymbol{U}^* \in \{-1,1\}^{k \times \ell}$. More visually, a binary matrix is $(k,\ell)$-biclustered if there exists some permutation of the rows and columns into a $k \times \ell$ grid of blocks each uniformly labeled $-1$ or $+1$, as illustrated in Figure 1. Determining if a matrix is in $\mathbb{B}_{k,\ell}^{m,n}$, may be done directly by a greedy algorithm. However, the problem of determining

if a matrix with missing entries may be completed to a matrix in $\mathbb{B}_{k,n}^{m,n}$ was shown in [29, Lemma 8] to be NP-COMPLETE by reducing the problem to CLIQUE COVER.

Many natural functions of matrix complexity are invariant to the presence of block structure. A function $f : \mathcal{X} \to \Re$ with respect to a class of matrices $\mathcal{X}$ is *block-invariant* if for all $m, k, n, \ell \in \mathbb{N}^+$ with $m \geq k$, $n \geq \ell$, $\boldsymbol{R} \in \mathcal{B}^{m,k}$ and $\boldsymbol{C} \in \mathcal{B}^{n,\ell}$ we have that $f(\boldsymbol{X}) = f(\boldsymbol{RXC}^\top)$ for any $k \times \ell$ matrix $\boldsymbol{X} \in \mathcal{X}$. The max-norm, margin complexity, rank and VC-dimension[2] are all block-invariant. From the block-invariance of the max-norm, we may conclude that for $\boldsymbol{U} \in \mathbb{B}_{k,\ell}^{m,n}$,

$$\mathrm{mc}(\boldsymbol{U}) \leq \|\boldsymbol{U}\|_{\max} = \|\boldsymbol{U}^*\|_{\max} \leq \min(\sqrt{k}, \sqrt{\ell}). \tag{7}$$

This follows since we may decompose $\boldsymbol{U} = \boldsymbol{R}\boldsymbol{U}^*\boldsymbol{C}^\top$ for some $\boldsymbol{U}^* \in \{-1, 1\}^{k \times \ell}$, $\boldsymbol{R} \in \mathcal{B}^{m,k}$ and $\boldsymbol{C} \in \mathcal{B}^{n,\ell}$ and then use the observation in the preliminaries that the max-norm of any matrix in $\{-1, 1\}^{m \times n}$ is bounded by $\min(\sqrt{m}, \sqrt{n})$.

In the following theorem, we give a bound for the quasi-dimension $\mathcal{D}_{\boldsymbol{M},\boldsymbol{N}}^\gamma(\boldsymbol{U})$ which will scale with the dimensions of the latent block structure and the "predictivity" of $\boldsymbol{M}$ and $\boldsymbol{N}$ with respect to that block structure. The bound is independent of $\gamma$ in so far as $\mathcal{D}_{\boldsymbol{M},\boldsymbol{N}}^\gamma(\boldsymbol{U})$ is finite.

**Theorem 3.** *If $\boldsymbol{U} \in \mathbb{B}_{k,\ell}^{m,n}$ define*

$$\mathcal{D}_{\boldsymbol{M},\boldsymbol{N}}^\circ(\boldsymbol{U}) := \begin{cases} 2\,\mathrm{tr}(\boldsymbol{R}^\top\boldsymbol{M}\boldsymbol{R})\mathcal{R}_{\boldsymbol{M}} + 2\,\mathrm{tr}(\boldsymbol{C}^\top\boldsymbol{N}\boldsymbol{C})\mathcal{R}_{\boldsymbol{N}} + 2k + 2\ell & \boldsymbol{M} \text{ and } \boldsymbol{N} \text{ are PDLaplacians} \\ k\,\mathrm{tr}(\boldsymbol{R}^\top\boldsymbol{M}\boldsymbol{R})\mathcal{R}_{\boldsymbol{M}} + \ell\,\mathrm{tr}(\boldsymbol{C}^\top\boldsymbol{N}\boldsymbol{C})\mathcal{R}_{\boldsymbol{N}} & \boldsymbol{M} \in \boldsymbol{S}_{++}^m \text{ and } \boldsymbol{N} \in \boldsymbol{S}_{++}^n \end{cases}, \tag{8}$$

*as the minimum over all decompositions of $\boldsymbol{U} = \boldsymbol{R}\boldsymbol{U}^*\boldsymbol{C}^\top$ for $\boldsymbol{R} \in \mathcal{B}^{m,k}$, $\boldsymbol{C} \in \mathcal{B}^{n,\ell}$ and $\boldsymbol{U}^* \in \{-1, 1\}^{k \times \ell}$. Thus for $\boldsymbol{U} \in \mathbb{B}_{k,\ell}^{m,n}$,*

$$\mathcal{D}_{\boldsymbol{M},\boldsymbol{N}}^\gamma(\boldsymbol{U}) \leq \mathcal{D}_{\boldsymbol{M},\boldsymbol{N}}^\circ(\boldsymbol{U}) \qquad (\text{if } \|\boldsymbol{U}\|_{max} \leq 1/\gamma)$$

$$\min_{\boldsymbol{V} \in \mathrm{SP}^1(\boldsymbol{U})} \mathcal{D}_{\boldsymbol{M},\boldsymbol{N}}^\gamma(\boldsymbol{V}) \leq \mathcal{D}_{\boldsymbol{M},\boldsymbol{N}}^\circ(\boldsymbol{U}) \qquad (\text{if } \mathrm{mc}(\boldsymbol{U}) \leq 1/\gamma).$$

The bound $\mathcal{D}_{\boldsymbol{M},\boldsymbol{N}}^\gamma(\boldsymbol{U}) \leq \mathcal{D}_{\boldsymbol{M},\boldsymbol{N}}^\circ(\boldsymbol{U})$ allows us to bound the quality of the side information in terms of a hypothetical learning problem. Recall that $\mathrm{argmin}_{r_i y_i \geq 1: i \in [m]}(\boldsymbol{r}^\top\boldsymbol{M}\boldsymbol{r})\mathcal{R}_{\boldsymbol{M}}$ is the upper bound on the mistakes per Novikoff's theorem [30] for predicting the elements of vector $\boldsymbol{y} \in \{-1, 1\}^m$ with a kernel perceptron using $\boldsymbol{M}^{-1}$ as the kernel. Hence the term $\mathcal{O}(\mathrm{tr}(\boldsymbol{R}^\top\boldsymbol{M}\boldsymbol{R})\mathcal{R}_{\boldsymbol{M}})$ in (8) may be interpreted as a bound for a one-versus-all $k$-class kernel perceptron where $\boldsymbol{R}$ encodes a labeling from $[k]^m$ as one-hot vectors. We next show an example where $\mathcal{D}_{\boldsymbol{M},\boldsymbol{N}}^\circ(\boldsymbol{U}) \in \mathcal{O}(k + \ell)$ with "ideal" side information.

### 4.1 Graph-based Side Information

We may use a pair of separate graph Laplacians to represent the side information on the "rows" and the "columns." A given row (column) corresponds to a vertex in the "row graph" ("column graph"). The weight of edge $(i, j)$ represents our prior belief that row (column) $i$ and row (column) $j$ share the same underlying factor. Such graphs may be inherent to the data. For example, we have a social network of users and a network based on shared actors or genres for the movies in a "Netflix" type scenario. Alternatively, as is common in graph-based semi-supervised learning [31; 32] we may build a graph based on vectorial data associated with the rows (columns), for example, user demographics. Although the value of $\mathcal{D}$ will vary smoothly with the predictivity of $\boldsymbol{M}$ and $\boldsymbol{N}$ of the factor structure, in the following we give an example to quantify $\mathcal{D}^\circ$ in a best case scenario.

**Bounding $\mathcal{D}^\circ$ for "ideal" graph-based side information**. In this ideal case we are assuming that we know the partition of $[m]$ that maps rows to factors. The rows that share factors have an edge between them and there are no other edges. Given $k$ factors, we then have a graph that consists of $k$ disjoint cliques. However, to meet the technical requirement that the side information matrix $\boldsymbol{M}(\boldsymbol{N})$ is positive definite, we need to connect the cliques in a minimal fashion. We achieve this by connecting the cliques like a "star" graph. Specifically, a clique is arbitrarily chosen as the center and a vertex in that clique is arbitrarily chosen as the central vertex. From each of the other cliques, a vertex is chosen arbitrarily and connected to the central vertex. Observe that a property of this construction is that there is a path of length $\leq 4$ between any pair of vertices. Now we can use the bound from Theorem 3,

$$\mathcal{D}^\circ = 2\,\mathrm{tr}(\boldsymbol{R}^\top\boldsymbol{M}\boldsymbol{R})\mathcal{R}_{\boldsymbol{M}} + 2\,\mathrm{tr}(\boldsymbol{C}^\top\boldsymbol{N}\boldsymbol{C})\mathcal{R}_{\boldsymbol{N}} + 2k + 2\ell,$$

to bound $\mathcal{D} \leq \mathcal{D}^\circ$ in this idealized case. We focus on the rows, as a parallel argument may be made for the side information on the columns. Consider the term $\mathrm{tr}(\boldsymbol{R}^\top \boldsymbol{M} \boldsymbol{R}) \mathcal{R}_M$, where $\boldsymbol{M} := \boldsymbol{L}^\circ$ is the PD-Laplacian formed from a graph with Laplacian $\boldsymbol{L}$. Then using the observation from the preliminaries that $(\boldsymbol{u}^\top \boldsymbol{L}^\circ \boldsymbol{u}) \mathcal{R}_{L^\circ} \leq 2(\boldsymbol{u}^\top \boldsymbol{L} \boldsymbol{u} \, \mathcal{R}_L + 1)$, we have that $\mathrm{tr}(\boldsymbol{R}^\top \boldsymbol{M} \boldsymbol{R}) \mathcal{R}_M \leq 2 \, \mathrm{tr}(\boldsymbol{R}^\top \boldsymbol{L} \boldsymbol{R}) \mathcal{R}_L + 2k$. To evaluate this, we use the well-known equality of $\mathrm{tr}(\boldsymbol{R}^\top \boldsymbol{L} \boldsymbol{R}) = \sum_{(i,j) \in E} \|\boldsymbol{R}_i - \boldsymbol{R}_j\|^2$. Observing that each of the $m$ rows of $\boldsymbol{R}$ is a "one-hot" encoding of the corresponding factor, only the edges between classes then contribute to the sum of the norms, and thus by construction $\mathrm{tr}(\boldsymbol{R}^\top \boldsymbol{L} \boldsymbol{R}) \leq k-1$. We bound $\mathcal{R}_L \leq 4$, using the fact that the graph diameter is a bound on $\mathcal{R}_L$ (see [33, Theorem 4.2]). Combining terms and assuming similar idealized side information on the columns, we obtain $\mathcal{D}^\circ \in O(k + \ell)$. Observe then that since the comparator matrix is $(k, \ell)$-biclustered, we have in the realizable case (with exact tuning), that $\mathrm{mc}(\boldsymbol{U})^2 \leq \min(k, \ell)$ by (7). Thus, the mistakes of the algorithm are bounded by $\tilde{\mathcal{O}}(\mathrm{mc}(\boldsymbol{U})^2 \mathcal{D}^\circ) = \tilde{\mathcal{O}}(k\ell)$. This upper bound is tight up to logarithmic factors as we may decompose $\boldsymbol{U} = \boldsymbol{R} \boldsymbol{U}^* \boldsymbol{C}^\top$ for some $\boldsymbol{U}^* \in \{-1, 1\}^{k \times \ell}$, $\boldsymbol{R} \in \mathcal{B}^{m,k}$ and $\boldsymbol{C} \in \mathcal{B}^{n,\ell}$ and force a mistake for each of the $k\ell$ entries in $\boldsymbol{U}^*$.

Can side information provably help? Unsurprisingly, yes. Consider the set of matrices such that each row is either all '+1' or all '-1'. This set is exactly $\mathbb{B}_{2,1}^{m,n}$. Clearly, an adversary can force $m$ mistakes, whereas with "ideal" side information the upper bound is $\tilde{\mathcal{O}}(1)$.

Similar results to the above can be obtained via alternate positive definite embeddings. For example, consider a *k-partition* kernel of $[m]$ where $K_{\epsilon, S_1, \ldots, S_k}(i, j) := [i, j \in S_r : r \in [k]] + \epsilon[i = j]$ for some partition of $[m]$ into disjoint sets $S_1, \ldots, S_k$. By using $\boldsymbol{M}^{-1} = (K(i, j))_{i,j \in [m]}$ one can obtain for small $\epsilon$, bounds that are tighter than achieved by the Laplacian with respect to constant factors. We have focused on the Laplacian as a method for encoding side information as it is more straightforward to encode [32] "softer" knowledge of relationships.

## 4.2 Online Community Membership Prediction

A special case of matrix completion is the case where there are $m$ objects which are assumed to lie in $k$ classes (communities). In this case, the underlying matrix $\boldsymbol{U} \in \{-1, 1\}$ is given by $U_{ij} = 1$ if $i$ and $j$ are in the same class and $U_{ij} = -1$ otherwise. Thus this may be viewed as an online version of community detection or "similarity" prediction. In [22], this problem was addressed when the side information was encoded in a graph and the aim was to perform well when there were few edges between classes (communities).

Observe that this is an example of a $(k, k)$-biclustered $m \times m$ matrix where $\boldsymbol{U}^* = 2\boldsymbol{I}^k - \boldsymbol{1}\boldsymbol{1}^\top$ and there exists $\boldsymbol{R} \in \mathcal{B}^{m,k}$ such that $\boldsymbol{U} := \boldsymbol{R} \boldsymbol{U}^* \boldsymbol{R}^\top$. Since the max-norm is block-invariant, we have that $\|\boldsymbol{U}\|_{\max} = \|\boldsymbol{U}^*\|_{\max}$. In the case of a general $k \times k$ biclustered matrix, $\|\boldsymbol{U}^*\|_{\max} \leq \sqrt{k}$ (see (7)). However in the case of "similarity prediction", we have $\|\boldsymbol{U}^*\|_{\max} \in O(1)$. This follows since we have a decomposition $\boldsymbol{U}^* = \boldsymbol{P}\boldsymbol{Q}^\top$ by $\boldsymbol{P}, \boldsymbol{Q} \in \Re^{k,k+1}$ with $\boldsymbol{P} := (P_{ij} = \sqrt{2}[i = j] + [j = k+1])_{i \in [k], j \in [k+1]}$ and $\boldsymbol{Q} := (Q_{ij} = \sqrt{2}[i = j] - [j = k+1])_{i \in [k], j \in [k+1]}$, thus giving $\|\boldsymbol{U}^*\|_{\max} \leq 3$. This example also shows that there may be an arbitrary gap between rank and max-norm of $\pm 1$ matrices as the rank of $\boldsymbol{U}^*$ is $k$ (in [6] this gap between the max-norm and rank was previously observed). Therefore, if the side-information matrices are taken to be the same PDLaplacian $\boldsymbol{M} = \boldsymbol{N}$ defined from a Laplacian $\boldsymbol{L}$, we have that since $\|\boldsymbol{U}\|_{\max} \in \mathcal{O}(1)$ and $\mathcal{D}^\circ \in \mathcal{O}(\mathrm{tr}(\boldsymbol{R}^\top \boldsymbol{L} \boldsymbol{R}) \mathcal{R}_L)$, a mistake bound of $\tilde{\mathcal{O}}(\mathrm{tr}(\boldsymbol{R}^\top \boldsymbol{L} \boldsymbol{R}) \mathcal{R}_L)$ is obtained, which recovers the bound of [22, Proposition 4] up to constant factors. This work extends the results in [22] for similarity prediction to regret bounds, and to the inductive setting with general p.d. matrices. In the next section, we will see how this type of result may be extended to an inductive setting.

## 5 Inductive Matrix Completion

In the previous section, the learner was assumed to have complete foreknowledge of the side information through the matrices $\boldsymbol{M}$ and $\boldsymbol{N}$. In the inductive setting, the learner has instead kernel side information functions $\mathcal{M}^+$ and $\mathcal{N}^+$. With complete foreknowledge of the rows (columns) that will be observed, one may use $\mathcal{M}^+$ ($\mathcal{N}^+$) to compute $\boldsymbol{M}$ ($\boldsymbol{N}$), which corresponds to an inverse of a submatrix of $\mathcal{M}^+$ ($\mathcal{N}^+$). In the inductive, unlike the transductive setting, we do not have this foreknowledge and thus cannot compute $\boldsymbol{M}$ ($\boldsymbol{N}$) in advance. Notice that the assumption of side information as kernel functions is not particularly limiting, as for instance the side information could be provided by vectors in $\Re^d$ and the kernel could be the positive definite linear kernel

$\boldsymbol{K}_\epsilon(\boldsymbol{x}, \boldsymbol{x}') := \langle \boldsymbol{x}, \boldsymbol{x}' \rangle + \epsilon[\boldsymbol{x} = \boldsymbol{x}']$. On the other hand, despite the additional flexibility of the inductive setting versus the transductive one, there are two limitations. First, only in a technical sense will it be possible to model side information via a PDLaplacian, since $\boldsymbol{M}^+$ can only be computed given knowledge of the graph in advance. Second, the bound in Theorem 3 on the quasi-dimension $\mathcal{D} \leq \mathcal{D}^\circ$ gains additional multiplicative factors $k$ and $\ell$. Nevertheless, we will observe in Section 5.1 that, for a given kernel for which the side information associated with a given row (column) latent factor is "well-separated" from distinct latent factors, we can show that $\mathcal{D}^\circ \in \mathcal{O}(k^2 + \ell^2)$.

---

**Algorithm 2** Predicting a binary matrix with side information in the inductive setting.

---

**Parameters:** Learning rate: $0 < \eta$ quasi-dimension estimate: $1 \leq \widehat{\mathcal{D}}$, margin estimate: $0 < \gamma \leq 1$, non-conservative flag [NON-CONSERVATIVE] $\in \{0, 1\}$ and side-information kernels $\mathcal{M}^+ : \mathcal{I} \times \mathcal{I} \to \Re$, $\mathcal{N}^+ : \mathcal{J} \times \mathcal{J} \to \Re$, with $\mathcal{R}_\mathcal{M} := \max_{i \in \mathcal{I}} \mathcal{M}^+(i, i)$ and $\mathcal{R}_\mathcal{N} := \max_{j \in \mathcal{J}} \mathcal{N}^+(j, j)$, and maximum distinct rows $m$ and columns $n$, where $m + n \geq 3$.

**Initialization:** $\mathbb{M} \leftarrow \emptyset$, $\mathbb{U} \leftarrow \emptyset$, $\mathcal{I}^1 \leftarrow \emptyset$, $\mathcal{J}^1 \leftarrow \emptyset$.

**For** $t = 1, \ldots, T$

- Receive pair $(i_t, j_t) \in \mathcal{I} \times \mathcal{J}$.
- Define
$$(\boldsymbol{M}^t)^+ := (\mathcal{M}^+(i_r, i_s))_{r,s \in \mathcal{I}^t \cup \{i_t\}}; \quad (\boldsymbol{N}^t)^+ := (\mathcal{N}^+(j_r, j_s))_{r,s \in \mathcal{J}^t \cup \{j_t\}},$$
$$\tilde{\boldsymbol{X}}^t(s) := \left[ \frac{(\sqrt{(\boldsymbol{M}^t)^+})\boldsymbol{e}^{i_s}}{\sqrt{2\mathcal{R}_\mathcal{M}}}; \frac{(\sqrt{(\boldsymbol{N}^t)^+})\boldsymbol{e}^{j_s}}{\sqrt{2\mathcal{R}_\mathcal{N}}} \right] \left[ \frac{(\sqrt{(\boldsymbol{M}^t)^+})\boldsymbol{e}^{i_s}}{\sqrt{2\mathcal{R}_\mathcal{M}}}; \frac{(\sqrt{(\boldsymbol{N}^t)^+})\boldsymbol{e}^{j_s}}{\sqrt{2\mathcal{R}_\mathcal{N}}} \right]^\top,$$
$$\log(\tilde{\boldsymbol{W}}^t) \leftarrow \log\left( \frac{\widehat{\mathcal{D}}}{m+n} \right) \boldsymbol{I}^{|\mathcal{I}^t|+|\mathcal{J}^t|+2} + \sum_{s \in \mathbb{U}} \eta y_s \tilde{\boldsymbol{X}}^t(s).$$
- Predict
$$Y_t \sim \text{UNIFORM}(-\gamma, \gamma) \times [\text{NON-CONSERVATIVE}]; \quad \bar{y}_t \leftarrow \text{tr}\left( \tilde{\boldsymbol{W}}^t \tilde{\boldsymbol{X}}^t \right) - 1; \quad \hat{y}_t \leftarrow \text{sign}(\bar{y}_t - Y_t).$$
- Receive label $y_t \in \{-1, 1\}$.
- If $y_t \neq \hat{y}_t$ then $\mathbb{M} \leftarrow \mathbb{M} \cup \{t\}$.
- If $y_t \bar{y}_t < \gamma \times [\text{NON-CONSERVATIVE}]$ then
$$\mathbb{U} \leftarrow \mathbb{U} \cup \{t\}, \quad \mathcal{I}^{t+1} \leftarrow \mathcal{I}^t \cup \{i_t\}, \text{ and } \mathcal{J}^{t+1} \leftarrow \mathcal{J}^t \cup \{j_t\}.$$
- Else $\mathcal{I}^{t+1} \leftarrow \mathcal{I}^t$ and $\mathcal{J}^{t+1} \leftarrow \mathcal{J}^t$.

---

Algorithm 2 is prediction-equivalent to Algorithm 1 up to the value of $\mathcal{R}_\mathcal{M}(\mathcal{R}_\mathcal{N})$. In [34], the authors provide very general conditions for the "kernelization" of algorithms with an emphasis on "matrix" algorithms. They sketch a method to kernelize the MATRIX EXPONENTIATED GRADIENT algorithm based on the relationship between the eigensystems of the kernel matrix and the Gram matrix. We take a different, more direct approach, in which we prove its correctness via Proposition 4. The intuition behind the algorithm is that, although we cannot efficiently embed the row and column kernel functions $\mathcal{M}^+$ and $\mathcal{N}^+$ as matrices since they are potentially infinite-dimensional, we may instead work with the embedding corresponding to the currently observed rows and columns, recompute the embedding on a per-trial basis, and then "replay" all re-embedded past examples to create the current hypothesis matrix. The following is our proposition of equivalency, proven in Appendix D.

**Proposition 4.** *The inductive and transductive algorithms are equivalent up to $\mathcal{R}_\mathcal{M}$ and $\mathcal{R}_\mathcal{N}$. Without loss of generality assume $\mathcal{I}^{T+1} \subseteq [m]$ and $\mathcal{J}^{T+1} \subseteq [n]$. Define $\boldsymbol{M} := ((\mathcal{M}^+(i', i''))_{i', i'' \in [m]})^+$ and $\boldsymbol{N} := ((\mathcal{N}^+(j', j''))_{j', j'' \in [n]})^+$. Assume that for the transductive algorithm, the matrices $\boldsymbol{M}$ and $\boldsymbol{N}$ are given whereas for the inductive algorithm, only the strictly positive definite kernel functions $\mathcal{M}^+$ and $\mathcal{N}^+$ are provided. Then, if $\mathcal{R}_\mathcal{M} = \mathcal{R}_{\boldsymbol{M}}$ and $\mathcal{R}_\mathcal{N} = \mathcal{R}_{\boldsymbol{N}}$, and if the algorithms receive the same label and index sequences, then the predictions of the algorithms are the same.*

Thus, the only case when the algorithms are different is when $\mathcal{R}_\mathcal{M} \neq \mathcal{R}_{\boldsymbol{M}}$ or $\mathcal{R}_\mathcal{N} \neq \mathcal{R}_{\boldsymbol{N}}$. This is a minor inequivalency, as the only resultant difference is in the term $\mathcal{D}$. Alternatively, if one uses a normalized kernel such as the Gaussian, then $\mathcal{R}_\mathcal{M} = \mathcal{R}_{\boldsymbol{M}} = 1$.

For Algorithm 1 we have a per trial time complexity of $\mathcal{O}(\max(m, n)^3)$; Algorithm 2 has a per trial complexity of $\mathcal{O}(\min(\max(m, n)^4, T^3))$. In the dominant step on every trial (with an update) of the transductive algorithm there is an SVD of a $(m + n) \times (m + n)$ matrix; thus, the algorithm requires $\mathcal{O}(\max(m, n)^3)$ time. We split the analysis of the inductive algorithm into two cases. In the case

that $\max(m, n) \ll T$, the complexity on every trial is dominated by the sum of up to $mn$ matrices of size up to $(m + n) \times (m + n)$ (i.e., in the regret setting we can collapse terms from multiple observations of the same matrix entry) and thus has a per-trial complexity of $\mathcal{O}(\max(m, n)^4)$. In the other case on trial $t$ we need $\mathcal{O}(t^3)$ time since we need to compute the eigendecomposition of three $\mathcal{O}(t) \times \mathcal{O}(t)$ matrices as well as sum $\mathcal{O}(t) \times \mathcal{O}(t)$ matrices up to $t$ times. Putting together we have a time complexity of $\mathcal{O}(\min(\max(m, n)^4, T^3))$ per trial. In the following subsection, we describe a scenario where the quasi-dimension bound $\mathcal{D}^\circ$ scales quadratically with the number of distinct factors.

## 5.1 Side information in $[-r, r]^d$

In the following, we show an example for predicting a matrix $\boldsymbol{U} \in \mathbb{B}_{k,\ell}^{m,n}$ such that for online side information in $[-r, r]^d$ that is well-separated into *boxes*, there exists a kernel for which the quasi-dimension grows no more than quadratically with the number of latent factors (but exponentially with the dimension $d$). For simplicity, we use the *min* kernel, which approximates functions by linear interpolation. In practice, we speculate that similar results may be proven for other universal kernels, but the analysis with the min kernel has the advantage of simplicity.

In the previous section, with the idealized graph-based side information, one may be dissatisfied as the skeleton of the latent structure is essentially encoded into $\boldsymbol{M}(\boldsymbol{N})$. In the inductive setting, the side information is instead revealed in an online fashion. If such side information may be separated into distinct clusters restrospectively, we will be able to bound $\mathcal{D}^\circ \in \mathcal{O}(k^2 + \ell^2)$. In this example, we receive a row and column vector $\imath_t, \jmath_t \in [-r, r]^d \times [-r', r']^{d'}$ on each trial; these vectors will be the indices to our row and column kernels, and for simplicity we set $r = r'$ and $d = d'$.

**Bounding $\mathcal{D}^\circ$ for the *min* kernel.** Define the transformation $s(\boldsymbol{x}) := \frac{r-1}{2r}\boldsymbol{x} + \frac{r+1}{2}$ and the *min* kernel $\mathcal{K} : [0, r]^d \times [0, r]^d \to \Re$ as $\mathcal{K}(\boldsymbol{x}, \boldsymbol{t}) := \prod_{i=1}^d \min(x_i, t_i)$. Also define $\delta(S_1, \ldots, S_k) := \min_{1 \le i < j \le k} \min_{\boldsymbol{x} \in S_i, \boldsymbol{x}' \in S_j} \|\boldsymbol{x} - \boldsymbol{x}'\|_\infty$. A *box* in $\Re^d$ is a set $\{\boldsymbol{x} : a_i \le x_i \le b_i, i \in [d]\}$ defined by a pair of vectors $\boldsymbol{a}, \boldsymbol{b} \in \Re^d$.

**Proposition 5.** *Given $k$ boxes $S_1, \ldots, S_k \subset [-r, r]^d$, $r \ge 2$, $\delta^* = \min\left(2, \frac{1}{4}\delta(S_1, \ldots, S_k)\right)$, and $\boldsymbol{x}_1, \ldots, \boldsymbol{x}_m \in \cup_{i=1}^k S_i$, if $\boldsymbol{R} = ([\boldsymbol{x}_i \in S_j])_{i \in [m], j \in [k]}$ and $\boldsymbol{K} = (\mathcal{K}(s(\boldsymbol{x}_i), s(\boldsymbol{x}_j)))_{i,j \in [m]}$ then $\operatorname{tr}(\boldsymbol{R}^\top \boldsymbol{K}^{-1} \boldsymbol{R}) \le k\left(\frac{4}{\delta^*}\right)^d$.*

Recall the bound (see (8)) on the quasi-dimension for a matrix $\boldsymbol{U} \in \mathbb{B}_{k,\ell}^{m,n}$, where we have $\mathcal{D} \le \mathcal{D}^\circ = k \operatorname{tr}(\boldsymbol{R}^\top \boldsymbol{M} \boldsymbol{R}) \mathcal{R}_{\boldsymbol{M}} + \ell \operatorname{tr}(\boldsymbol{C}^\top \boldsymbol{N} \boldsymbol{C}) \mathcal{R}_{\boldsymbol{N}}$ for positive definite matrices. If we assume that the side information on the rows (columns) lies in $[-r, r]^d$, then $\mathcal{R}_{\boldsymbol{M}} \le \mathcal{R}_{\mathcal{M}} \le r^d$ ($\mathcal{R}_{\boldsymbol{N}} \le \mathcal{R}_{\mathcal{N}} \le r^d$) for the min kernel. Thus by applying the above proposition separately for the rows and columns and substituting into (8), we have that

$$\mathcal{D} \le \mathcal{D}^\circ = k^2 (4r/\delta^*)^d + \ell^2 (4r/\delta^*)^d.$$

We then observe that for this example, with an optimal tuning and well-separated side information on the rows and columns, the mistake bound for a $(k, \ell)$-biclustered matrix in the inductive setting is of $\tilde{\mathcal{O}}(\min(k, \ell) \max(k, \ell)^2)$. However, our best lower bound in terms of $k$ and $\ell$ is just $k\ell$, as in the transductive setting. An open problem is to resolve this gap.

## 6 Discussion

We have presented a regret bound with respect to the 0-1 loss for predicting the elements of matrices with latent block structure in the presence of side information. The bound scales as $\tilde{\mathcal{O}}\left(\sqrt{\mathcal{D}_{\boldsymbol{M}, \boldsymbol{N}}^\gamma(\boldsymbol{U}) \min(k, \ell) T}\right)$ for $\boldsymbol{U} \in \mathbb{B}_{k,\ell}^{m,n}$. In the case of idealized side information, the term $\mathcal{D}$ scaled linearly with $k$ and $\ell$ in the transductive setting, and quadratically in the inductive setting. Problems for further research include resolving the gap between the upper and lower bounds in the inductive setting and building richer models of side information. In this work, the interrelation was restricted to equivalence relations on the row (column) spaces. A direct generalization would be to consider partial orderings. Such a generalization would be natural for ranking based on concept classes such as the online gambling scenario given in [21]. A broader generalization would allow side information to hint between row and column interrelationships rather than keeping them independent.

# 7 Acknowledgements

We would like to thank Robin Hirsch for valuable discussions. This research was supported by the U.S. Army Research Laboratory and the U.K. Ministry of Defence under Agreement Number W911NF-16-3-0001. The views and conclusions contained in this document are those of the authors and should not be interpreted as representing the official policies, either expressed or implied, of the U.S. Army Research Laboratory, the U.S. Government, the U.K. Ministry of Defence or the U.K. Government. The U.S. and U.K. Governments are authorized to reproduce and distribute reprints for Government purposes notwithstanding any copyright notation hereon. This research was further supported by the Engineering and Physical Sciences Research Council grant number EP/L015242/1.

## Broader Impact

*In general this work does not present any foreseeable specific societal consequence in the authors' joint opinion.*

This is foundational research in *regret-bounded online learning*. As such it is not targeted towards any particular application area. Although this research may have societal impact for good or for ill in the future, we cannot foresee the shape and the extent.

## Funding Transparency Statement

### Funding

The authors were supported by the U.S. Army Research Laboratory and the U.K. Ministry of Defence under agreement number W911NF-16-3-0001 and by the Engineering and Physical Sciences Research Council grant number EP/L015242/1.

### Competing Interests

The authors assert no competing interests.

## Footnotes

[1] In [21], a regret bound for general loss functions for matrix completion without side information is given for $(\beta, \tau)$-decomposable matrices. When $\beta$ is at its minimum over all possible decompositions, we recover the bound up to constant factors with respect to the expected 0-1 loss. On the algorithmic level, our works are similar except that the algorithm of [21] contains an additional projection step that dominates the computation time of the update.

[2]Here, a hypothesis class $\mathcal{H}$ defines a matrix via $\boldsymbol{U} := (h(x))_{h \in \mathcal{H}, x \in \mathcal{X}}$.

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
