[Supplementary Material]

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

# A  Synthetic Experiments

To illustrate the algorithm's performance, synthetic experiments were performed in the transductive setting with graph side information. In particular, we took $U$ to be randomly generated square (9,9)-biclustered matrices with i.i.d. noise. A visualization of a noise-free example matrix can be found in Figure 1. The noise process flipped the label of each matrix entry independently with probability $p = 0.10$. The side information on the rows and columns were represented by PDLaplacian matrices, for which the underlying graphs were constructed in the manner described in Section 4.1. Varying levels of side information noise $\beta \in [0.0, 0.5]$ were applied. This was introduced by considering every pair of vertices independently from the constructed graph and flipping the state between EDGE/NOT-EDGE with probability $\beta$. A final step is added to ensure the graph is connected. In this step a random pair of components is connected by a random edge, recursively. The process terminates when the graph is connected.

Figure 2: Error rates for predicting a noisy $(9, 9)$-biclustered matrix with side information.

The parameters were chosen so that the expected regret bound in Theorem 1 would apply to our experimental setting. We use the quasi-dimension upper bound $\widehat{\mathcal{D}} := \mathcal{D}^{\circ}_{M,N}(U) = 2\operatorname{tr}(R^{\top}MR)\mathcal{R}_M + 2\operatorname{tr}(C^{\top}NC)\mathcal{R}_N + 4k$, as developed in Theorem 3 for PDLaplacians. The learning rate was set as $\eta = \sqrt{\frac{\widehat{\mathcal{D}}\log(2n)}{2T}}$. Since each run of the algorithm consisted of predicting all $n^2$ matrix entries sampled uniformly at random without replacement, we set $T = n^2$. As for the margin estimate, due to the requirement that $\gamma \leq 1/\|U\|_{\max}$, a suitable value can be extracted from Equation (7), giving $\gamma = 1/\sqrt{k}$.

The per trial mistake rate is shown in Fig. 2 for matrix dimension $n = 20, \ldots, 400$, where each data point is averaged over 10 runs. We observe that for random side information $\beta = 0.5$, the term $\widehat{\mathcal{D}}$ could lead to a bound which is vacuous (for small $n$), however, the algorithm's error rate was in the range of $[0.30, 0.45]$, being well below chance. With ideal side information, $\beta = 0.0$, the performance improved drastically, as suggested by the bounds, to an error rate in $[0.10, 0.35]$. Observe that since there is 10% label noise for all values of $\beta$, the curves are converging to an online mistake rate of 10%. The data points for the plot can be found in Table 1. For our specific implementation of $n = 400$, each run required approximately 6 hours on a 3.40GHz Intel(R) Xeon(R) CPU E3-1240 v3 with 4GB of RAM. Note that this was run on a shared node on a cluster, meaning that the run time may vary depending on the other jobs that were running.

# B  Proof of Theorem 1

The proof of Theorem 1 is organized as follows. We start with the required preliminaries in Subsection B.1, and then proceed to prove the regret statement of the theorem, given by Equation (5), in Subsection B.2. Finally, in Subsection B.3, we provide a proof for the mistake bound in the realizable case, as stated in Equation (6).

## B.1  Preliminaries for Proof

Suppose we have $M$, $N$ and $U$ as in Theorem 1. Instead of working with $U$ directly, we shall work with an embedding of this matrix. We have different treatments for the embedding of $U$ in the two parts of the proof. For Subsection B.2, let $\bar{U} \in \Re^{m \times n}$ be such that $\bar{U} = \gamma U$. Following from the assumption that $\|U\|_{\max} \leq \frac{1}{\gamma}$, there exist row-normalized matrices $\hat{P} \in \Re^{n \times d}$ and $\hat{Q} \in \Re^{m \times d}$ that give $\bar{U} = \hat{P}\hat{Q}^{\top}$. In Subsection B.3, however, we have that $\bar{U} = \gamma \operatorname*{argmin}_{V \in \mathrm{SP}^1(U)} \mathcal{D}^{\gamma}_{M,N}(V)$.

Table 1: Data points used for Fig. 2.

| Matrix Dimensions $n$ | Noise | | | | | |
|---|---|---|---|---|---|---|
| | 0.50 | 0.25 | 0.125 | 0.0625 | 0.03125 | 0.00 |
| 20 | 0.39± 0.04 | 0.4± 0.05 | 0.38± 0.04 | 0.34± 0.04 | 0.31± 0.03 | 0.31± 0.03 |
| 40 | 0.37± 0.03 | 0.38± 0.03 | 0.34± 0.02 | 0.33± 0.02 | 0.29± 0.02 | 0.22± 0.01 |
| 60 | 0.37± 0.02 | 0.35± 0.02 | 0.34± 0.02 | 0.32± 0.02 | 0.27± 0.02 | 0.18± 0.01 |
| 80 | 0.36± 0.02 | 0.34± 0.02 | 0.33± 0.01 | 0.29± 0.02 | 0.25± 0.02 | 0.16± 0.01 |
| 100 | 0.36± 0.02 | 0.35± 0.02 | 0.33± 0.01 | 0.29± 0.01 | 0.24± 0.01 | 0.15± 0.01 |
| 120 | 0.35± 0.02 | 0.34± 0.02 | 0.31± 0.01 | 0.29± 0.01 | 0.24± 0.02 | 0.14± 0.01 |
| 140 | 0.35± 0.02 | 0.34± 0.01 | 0.31± 0.01 | 0.28± 0.01 | 0.22± 0.01 | 0.13± 0.01 |
| 160 | 0.35± 0.02 | 0.33± 0.02 | 0.31± 0.02 | 0.27± 0.01 | 0.22± 0.01 | 0.13± 0.0 |
| 180 | 0.34± 0.02 | 0.33± 0.02 | 0.3± 0.02 | 0.26± 0.01 | 0.22± 0.01 | 0.13± 0.0 |
| 200 | 0.33± 0.02 | 0.32± 0.02 | 0.29± 0.01 | 0.26± 0.01 | 0.21± 0.01 | 0.13± 0.0 |
| 250 | 0.32± 0.02 | 0.31± 0.02 | 0.29± 0.01 | 0.24± 0.01 | 0.21± 0.01 | 0.12± 0.0 |
| 300 | 0.32± 0.02 | 0.31± 0.01 | 0.28± 0.01 | 0.23± 0.01 | 0.2± 0.01 | 0.12± 0.0 |
| 400 | 0.3± 0.01 | 0.28± 0.02 | 0.26± 0.01 | 0.23± 0.01 | 0.19± 0.01 | 0.11± 0.0 |

In this section we assume $\mathrm{mc}(U) \leq \frac{1}{\gamma}$, which also guarantees a decomposition in terms of the row-normalized matrices $\hat{P}$ and $\hat{Q}$ such that $\bar{U} = \hat{P}\hat{Q}^\top$.

Let us then define the quasi-dimension with respect to a specific factorization

$$\mathcal{D} := \mathcal{R}_M \, \mathrm{tr}\left(\hat{P}^\top M \hat{P}\right) + \mathcal{R}_N \, \mathrm{tr}\left(\hat{Q}^\top N \hat{Q}\right).$$

Note that in general for the case that $\bar{U} = \gamma U$, $\mathcal{D} \geq \mathcal{D}^\gamma_{M,N}(U)$, whereas for the case that $\bar{U} \in \gamma\mathrm{SP}^1(U)$, $\mathcal{D} \geq \min_{V \in \mathrm{SP}^1(U)} \mathcal{D}^\gamma_{M,N}(V)$. We proceed with the proof assuming that $(\hat{P}, \hat{Q})$ is the optimal factorization for a given $\bar{U}$. That is, for $\bar{U} = \gamma U$, we have that $(\hat{P}, \hat{Q})$ is the factorization that satisfies $\mathcal{D} = \mathcal{D}^\gamma_{M,N}(U)$, and for $\bar{U} = \gamma \, \mathrm{argmin}_{V \in \mathrm{SP}^1(U)} \, \mathcal{D}^\gamma_{M,N}(V)$, $(\hat{P}, \hat{Q})$ satisfies $\mathcal{D} = \min_{V \in \mathrm{SP}^1(U)} \mathcal{D}^\gamma_{M,N}(V)$.

Next, we define $\tilde{U}$, which is a positive semidefinite matrix, used as an embedding for $\bar{U}$ in the analysis of the algorithm. Its exact relationship with $\bar{U}$ is shown in Lemma 7.

**Definition 6.** *Define the $(m+n) \times (m+n)$ matrix $Z$ as*

$$Z := \begin{pmatrix} \sqrt{\mathcal{R}_M}\sqrt{M}\hat{P} \\ \sqrt{\mathcal{R}_N}\sqrt{N}\hat{Q} \end{pmatrix}. \tag{9}$$

*and construct $\tilde{U}$ as,*

$$\tilde{U} := ZZ^\top = \begin{pmatrix} \mathcal{R}_M\sqrt{M}\hat{P}\hat{P}^\top\sqrt{M} & \sqrt{\mathcal{R}_M\mathcal{R}_N}\sqrt{M}\hat{P}\hat{Q}^\top\sqrt{N} \\ \sqrt{\mathcal{R}_M\mathcal{R}_N}\sqrt{N}\hat{Q}\hat{P}^\top\sqrt{M} & \mathcal{R}_N\sqrt{N}\hat{Q}\hat{Q}^\top\sqrt{N} \end{pmatrix}.$$

**Lemma 7.** *For all trials $t \in [T]$,*

$$\bar{U}_{i_t j_t} = \mathrm{tr}\left(\tilde{U}\tilde{X}^t\right) - 1$$

*where $\tilde{U}$ is as constructed from Definition 6.*

*Proof.* We have:

$$\mathrm{tr}\left(\tilde{U}\tilde{X}^t\right) = \left(x^t\right)^\top \tilde{U} x^t \tag{10}$$

$$= (x^t)^\top ZZ^\top x^t \tag{11}$$

$$= \left\| (x^t)^\top Z \right\|^2 . \tag{12}$$

Recall that

$$x^t = \left[ \frac{\sqrt{M^+} e_m^{i_t}}{\sqrt{2\mathcal{R}_M}} ; \frac{\sqrt{N^+} e_n^{j_t}}{\sqrt{2\mathcal{R}_N}} \right] \text{ and } Z = \begin{pmatrix} \sqrt{\mathcal{R}_M}\sqrt{M}\hat{P} \\ \sqrt{\mathcal{R}_N}\sqrt{N}\hat{Q} \end{pmatrix}$$

Hence,

$$(x^t)^\top Z = \frac{(\sqrt{M^+} e_m^{i_t})^\top}{\sqrt{2\mathcal{R}_M}} \sqrt{\mathcal{R}_M}\sqrt{M}\hat{P} + \frac{(\sqrt{N^+} e_n^{j_t})^\top}{\sqrt{2\mathcal{R}_N}} \sqrt{\mathcal{R}_N}\sqrt{N}\hat{Q}$$

$$= \frac{1}{\sqrt{2}} (e_m^{i_t})^\top \sqrt{M^+}\sqrt{M}\hat{P} + \frac{1}{\sqrt{2}} (e_n^{j_t})^\top \sqrt{N^+}\sqrt{N}\hat{Q}$$

$$= \frac{1}{\sqrt{2}} (\hat{P}_{i_t} + \hat{Q}_{j_t}) \tag{13}$$

Thus substituting (13) into (12) gives,

$$\mathrm{tr}\left( \tilde{U}\tilde{X}^t \right) = \frac{1}{2} \left\| \hat{P}_{i_t} + \hat{Q}_{j_t} \right\|^2$$

$$= \frac{1}{2} \left( \left\| \hat{P}_{i_t} \right\|^2 + 2\langle \hat{P}_i, \hat{Q}_{j_t}\rangle + \left\| \hat{Q}_{j_t} \right\|^2 \right)$$

$$= \left( 1 + \langle \hat{P}_{i_t}, \hat{Q}_{j_t}\rangle \right)$$

$$= 1 + \bar{U}_{i_t j_t} .$$

$$\square$$

In the subsequent proofs, we will also need to make use of the following facts.

**Lemma 8.** *For $\tilde{U}$ as defined in Definition 6, we have that,*

$$\mathrm{tr}(\tilde{U}) = \mathcal{D} . \tag{14}$$

*Proof.*

$$\mathrm{tr}(\tilde{U}) = \mathrm{tr}(ZZ^\top) = \mathrm{tr}\left( \begin{pmatrix} \sqrt{\mathcal{R}_M}\sqrt{M}\hat{P} \\ \sqrt{\mathcal{R}_N}\sqrt{N}\hat{Q} \end{pmatrix} \begin{pmatrix} \sqrt{\mathcal{R}_M}\sqrt{M}\hat{P} & \sqrt{\mathcal{R}_N}\sqrt{N}\hat{Q} \end{pmatrix}^\top \right)$$

$$= \mathcal{R}_M \mathrm{tr}\left( \sqrt{M}\hat{P}\hat{P}^\top \sqrt{M}^\top \right) + \mathcal{R}_N \mathrm{tr}\left( \sqrt{N}\hat{Q}\hat{Q}^\top \sqrt{N}^\top \right)$$

$$= \mathcal{R}_M \mathrm{tr}\left( \hat{P}^\top M\hat{P} \right) + \mathcal{R}_N \mathrm{tr}\left( \hat{Q}^\top N\hat{Q} \right)$$

$$= \mathcal{D}$$

$$\square$$

**Lemma 9.** *For all trials $t$, all eigenvalues of $\tilde{X}^t$ are in $[0,1]$.*

*Proof.* Recall that

$$\mathrm{tr}(\tilde{X}^t) = \mathrm{tr}(x^t (x^t)^\top) = \left[ \frac{\sqrt{M^+} e_m^{i_t}}{\sqrt{2\mathcal{R}_M}} ; \frac{\sqrt{N^+} e_n^{j_t}}{\sqrt{2\mathcal{R}_N}} \right]^\top \left[ \frac{\sqrt{M^+} e_m^{i_t}}{\sqrt{2\mathcal{R}_M}} ; \frac{\sqrt{N^+} e_n^{j_t}}{\sqrt{2\mathcal{R}_N}} \right] .$$

Hence

$$\left\| x^t \right\|^2 = \left\| \frac{\sqrt{M^+} e_m^{i_t}}{\sqrt{2\mathcal{R}_M}} \right\|^2 + \left\| \frac{\sqrt{N^+} e_n^{j_t}}{\sqrt{2\mathcal{R}_N}} \right\|^2$$

and then bounding the first term on the right hand side gives,

$$\left\| \frac{\sqrt{M^+} e_m^{i_t}}{\sqrt{2\mathcal{R}_M}} \right\|^2 = \frac{1}{2\mathcal{R}_M} \left( e_m^{i_t} \right)^\top \left( \sqrt{M^+} \right)^\top \sqrt{M^+} e_m^{i_t} \leq \frac{1}{2\mathcal{R}_M} \max_{i \in [m]} \left( e_m^{i_t} \right)^\top M^+ e_m^{i_t} = \frac{1}{2}.$$

The argument for the second term is parallel. Therefore since it is shown that the trace of $\tilde{X}^t$ is bounded by 1 and that $\tilde{X}^t$ is positive definite, this implies that all eigenvalues of $\tilde{X}^t$ are in $[0,1]$. □

Next, we introduce the following quantity, which plays a central role in the amortized analysis of our algorithm.

**Definition 10.** *The quantum relative entropy of symmetric positive semidefinite square matrices $A$ and $B$ is*

$$\Delta(A, B) := \mathrm{tr}(A \log(A) - A \log(B) + B - A).$$

An important result that will be used in the subsequent subsections is the well known Golden-Thompson Inequality, whose proof can be found, for example, in [35].

**Lemma 11.** *For any symmetric matrices $A$ and $B$ we have,*

$$\mathrm{tr}(\exp(A + B)) \leq \mathrm{tr}(\exp(A) \exp(B)).$$

## B.2   Proof for the Regret Statement

In this subsection, we prove the regret bound as presented in Theorem 1, which holds for Algorithm 1 with **non-conservative** updates. To do so, we first derive a regret bound in terms of the hinge loss with the deterministic prediction $\bar{y}_t$. We then convert this to an expected regret bound in terms of the zero-one loss for the random variable $\hat{y}_t$.

Regret bounds for the MEG algorithm were originally proven in [25]. However, that analysis leads to a $\mathrm{tr}(\tilde{U})$ dependence, whereas we derive a $\sqrt{\mathrm{tr}(\tilde{U})}$ scaling, for our more restrictive setting. Regret bounds with such scaling for linear classification in the vector case have been previously given in [27] (which themselves are generalisations of the bounds from Littlestone [26] for learning $k$-literal disjunctions with $\mathcal{O}(k \log n)$ mistakes). However, to our knowledge, no such regret bounds for MEG are present in the literature for the matrix case. Our proof uses an amortized analysis of the quantum relative entropy, followed by an application of the matricized results in [27]. The original results in [27] are built on the results in [36], which use convex arguments. We eliminate the need to introduce convex concepts to our proof by instead using an amortized analysis. We conclude the proof by using algebraic arguments to transform the regret in terms of the hinge loss to an expected regret in terms of the zero-one loss.

In the following, we define the hinge loss as $h_\gamma(y, \bar{y}) := \frac{1}{\gamma}[\gamma - y\bar{y}]_+$. We define

$$H^t := \nabla_{\tilde{W}} \gamma h_\gamma(y_t, \bar{y}_t), \tag{15}$$

where $\nabla$ denotes the subgradient and where $\bar{y}_t$ is as defined in Algorithm 1. When $y_t \bar{y}_t = \gamma$, we will only consider the specific subgradient $H^t = 0$.

**Lemma 12.** *For all $t \in [T]$,*

$$H^t = -y_t \tilde{X}^t \left[ \gamma > y_t \left( \mathrm{tr}\left( \tilde{W}^t \tilde{X}^t \right) - 1 \right) \right].$$

*Proof.* Recalling the definition of $H^t := \nabla_{\tilde{W}} \gamma h_\gamma(y_t, \bar{y}_t)$, observe that when $\gamma > y_t \left( \mathrm{tr}\left( \tilde{W}^t \tilde{X}^t \right) - 1 \right)$, we have

$$\nabla_{\tilde{W}} \gamma h_\gamma(y_t, \bar{y}_t) = \nabla_{\tilde{W}} \left[ \gamma - y_t \left( \mathrm{tr}\left( \tilde{W}^t \tilde{X}^t \right) - 1 \right) \right]_+ = -y_t (\tilde{X}^t)^\top = -y_t \tilde{X}^t, \tag{16}$$

where we used the fact that $\nabla_A \mathrm{tr}\left( AB \right) = B^\top$. In the case that $\gamma \leq y_t \left( \mathrm{tr}\left( \tilde{W}^t \tilde{X}^t \right) - 1 \right)$,

$$\nabla_{\tilde{W}} \gamma h_\gamma(y_t, \bar{y}_t) = 0.$$

□

**Lemma 13.** *For matrix $\boldsymbol{A} \in \boldsymbol{S}^d$ with eigenvalues no less than -1,*

$$\boldsymbol{I} - \boldsymbol{A} + \boldsymbol{A}^2 - \exp(-\boldsymbol{A}) \succeq \boldsymbol{0}.$$

*Proof.* Let $\boldsymbol{B} := \boldsymbol{I} - \boldsymbol{A} + \boldsymbol{A}^2 - \exp(-\boldsymbol{A})$. Observing that $\boldsymbol{A}$, $\boldsymbol{A}^2$ and $\exp(-\boldsymbol{A})$ share the same set of eigenvectors,

$$\boldsymbol{I} - \boldsymbol{A} + \boldsymbol{A}^2 - \exp(-\boldsymbol{A}) = \boldsymbol{U}(\boldsymbol{I} - \boldsymbol{\Lambda} + \boldsymbol{\Lambda}^2 - \exp(-\boldsymbol{\Lambda}))\boldsymbol{U}^\top,$$

where $\boldsymbol{U}$ is the orthogonal matrix and $\boldsymbol{\Lambda}$ is the diagonal matrix in the eigendecomposition of $\boldsymbol{A}$. Therefore, each eigenvalue $\lambda_{\boldsymbol{B},i}$ of the resulting matrix $\boldsymbol{B}$ can be written in terms of an eigenvalue $\lambda_{\boldsymbol{B},i}$ of matrix $\boldsymbol{A}$ for all $i \in [d]$,

$$\lambda_{\boldsymbol{B},i} = 1 - \lambda_{\boldsymbol{A},i} + \lambda_{\boldsymbol{A},i}^2 - \exp(-\lambda_{\boldsymbol{A},i}).$$

**Lemma 14.** *For any matrix $\boldsymbol{A} \in \boldsymbol{S}_{++}^d$ and any two matrices $\boldsymbol{B}, \boldsymbol{C} \in \boldsymbol{S}^d$, $\boldsymbol{B} \preceq \boldsymbol{C}$ implies* $\operatorname{tr}(\boldsymbol{AB}) \leq \operatorname{tr}(\boldsymbol{AC})$.

*Proof.* This follows a parallel argument to the proof for [25, Lemma 2.2]. □

The positive semidefinite criterion requires that all eigenvalues be non-negative, so that $\lambda_{\boldsymbol{B},i} \geq 0$. This inequality holds true for $\lambda_{\boldsymbol{A},i} \geq -1$. □

**Lemma 15.** *For all trials $t \in [T]$ in Algorithm 1, we have for $\eta \in (0,1]$:*

$$\Delta(\tilde{\boldsymbol{U}}, \tilde{\boldsymbol{W}}^t) - \Delta(\tilde{\boldsymbol{U}}, \tilde{\boldsymbol{W}}^{t+1}) \geq \eta \left( \operatorname{tr}((\tilde{\boldsymbol{W}}^t - \tilde{\boldsymbol{U}})\boldsymbol{H}^t) - \eta^2 \operatorname{tr}(\tilde{\boldsymbol{W}}^t(\boldsymbol{H}^t)^2) \right). \tag{17}$$

*Proof.* We have:

$$\begin{aligned}
\Delta(\tilde{\boldsymbol{U}}, \tilde{\boldsymbol{W}}^t) - \Delta(\tilde{\boldsymbol{U}}, \tilde{\boldsymbol{W}}^{t+1}) &= \operatorname{tr}\left(\tilde{\boldsymbol{U}}\log\tilde{\boldsymbol{W}}^{t+1} - \tilde{\boldsymbol{U}}\log\tilde{\boldsymbol{W}}^t\right) + \operatorname{tr}\left(\tilde{\boldsymbol{W}}^t\right) - \operatorname{tr}\left(\tilde{\boldsymbol{W}}^{t+1}\right) \\
&= -\eta\operatorname{tr}\left(\tilde{\boldsymbol{U}}\boldsymbol{H}^t\right) + \operatorname{tr}\left(\tilde{\boldsymbol{W}}^t\right) - \operatorname{tr}\left(e^{\log\tilde{\boldsymbol{W}}^t - \eta\boldsymbol{H}^t}\right) \tag{18} \\
&\geq -\eta\operatorname{tr}\left(\tilde{\boldsymbol{U}}\boldsymbol{H}^t\right) + \operatorname{tr}\left(\tilde{\boldsymbol{W}}^t\right) - \operatorname{tr}\left(e^{\log\tilde{\boldsymbol{W}}^t}e^{-\eta\boldsymbol{H}^t}\right) \tag{19} \\
&= -\eta\operatorname{tr}\left(\tilde{\boldsymbol{U}}\boldsymbol{H}^t\right) + \operatorname{tr}\left(\tilde{\boldsymbol{W}}^t\left(\boldsymbol{I} - e^{-\eta\boldsymbol{H}^t}\right)\right) \\
&\geq -\eta\operatorname{tr}\left(\tilde{\boldsymbol{U}}\boldsymbol{H}^t\right) + \operatorname{tr}\left(\tilde{\boldsymbol{W}}^t\left(\eta\boldsymbol{H}^t - \eta^2(\boldsymbol{H}^t)^2\right)\right) \tag{20}
\end{aligned}$$

where Equation (18) comes from the update of the algorithm and Lemma 12, Equation (19) comes from Lemma 11 and Equation (20) comes from Lemmas 13 and 14. □

**Lemma 16.** *For Algorithm 1, we have for $\eta \in (0,1]$:*

$$\sum_{t=1}^T \operatorname{tr}\left((\tilde{\boldsymbol{W}}^t - \tilde{\boldsymbol{U}})\boldsymbol{H}^t\right) \leq \frac{1}{\eta}\left(\operatorname{tr}\left(\tilde{\boldsymbol{U}}\log\left(\frac{\tilde{\boldsymbol{U}}(m+n)}{e\widehat{\mathcal{D}}}\right)\right) + \widehat{\mathcal{D}}\right) + \sum_{t=1}^T \eta\operatorname{tr}\left(\tilde{\boldsymbol{W}}^t(\boldsymbol{H}^t)^2\right). \tag{21}$$

*Setting the additional assumptions $\widehat{\mathcal{D}} \geq \operatorname{tr}\left(\tilde{\boldsymbol{U}}\right) \geq 1$ and $m+n \geq 3$ gives*

$$\sum_{t=1}^T \operatorname{tr}\left((\tilde{\boldsymbol{W}}^t - \tilde{\boldsymbol{U}})\boldsymbol{H}^t\right) \leq \frac{\widehat{\mathcal{D}}}{\eta}\log(m+n) + \sum_{t=1}^T \eta\operatorname{tr}\left(\tilde{\boldsymbol{W}}^t(\boldsymbol{H}^t)^2\right). \tag{22}$$

*Proof.* We start by proving Equation (21). Rearranging Lemma 15 and summing over $t$,

$$\begin{aligned}
\sum_{t=1}^T \left(\operatorname{tr}((\tilde{\boldsymbol{W}}^t - \tilde{\boldsymbol{U}})\boldsymbol{H}^t)\right) &\leq \frac{1}{\eta}\left(\sum_{t=1}^T \Delta(\tilde{\boldsymbol{U}}, \tilde{\boldsymbol{W}}^t) - \Delta(\tilde{\boldsymbol{U}}, \tilde{\boldsymbol{W}}^{t+1}) + \eta^2\operatorname{tr}\left(\tilde{\boldsymbol{W}}^t(\boldsymbol{H}^t)^2\right)\right) \\
&\leq \frac{1}{\eta}\left(\Delta(\tilde{\boldsymbol{U}}, \tilde{\boldsymbol{W}}^1) - \Delta(\tilde{\boldsymbol{U}}, \tilde{\boldsymbol{W}}^{T+1}) + \sum_{t=1}^T \eta^2\operatorname{tr}\left(\tilde{\boldsymbol{W}}^t(\boldsymbol{H}^t)^2\right)\right).
\end{aligned}$$

Using the fact that $\Delta(\tilde{U}, \tilde{W}^{T+1}) \geq 0$ and writing out $\Delta(\tilde{U}, \tilde{W}^1)$, we then obtain Equation (21).

To prove Equation (22), we attempt to maximize the term $\operatorname{tr}\left(\tilde{U} \log\left(\frac{\tilde{U}(m+n)}{e\widehat{\mathcal{D}}}\right)\right) + \widehat{\mathcal{D}}$. Noting that $\operatorname{tr}\left(\tilde{U} \log(a\tilde{U})\right) \leq \operatorname{tr}(\tilde{U}) \log\left(\operatorname{tr}(a\tilde{U})\right)$ for $a \geq 0$, we then have

$$\operatorname{tr}\left(\tilde{U} \log\left(\frac{\tilde{U}(m+n)}{e\widehat{\mathcal{D}}}\right)\right) + \widehat{\mathcal{D}} \leq \operatorname{tr}(\tilde{U}) \log\left(\frac{\operatorname{tr}(\tilde{U})(m+n)}{e\widehat{\mathcal{D}}}\right) + \widehat{\mathcal{D}}.$$

The upper bound in the above equation is convex in $\operatorname{tr}(\tilde{U})$ and hence is maximized at either boundary $\{1, \widehat{\mathcal{D}}\}$. Comparing the terms, we have

$$\operatorname{tr}\left(\tilde{U} \log\left(\frac{\tilde{U}(m+n)}{e\widehat{\mathcal{D}}}\right)\right) + \widehat{\mathcal{D}} \leq \log\left(\frac{m+n}{e\widehat{\mathcal{D}}}\right) + \widehat{\mathcal{D}}$$

for $\operatorname{tr}(\tilde{U}) = 1$ and

$$\operatorname{tr}\left(\tilde{U} \log\left(\frac{\tilde{U}(m+n)}{e\widehat{\mathcal{D}}}\right)\right) + \widehat{\mathcal{D}} \leq \widehat{\mathcal{D}} \log(m+n)$$

for $\operatorname{tr}(\tilde{U}) = \widehat{\mathcal{D}}$. We then observe that given the assumptions, $\widehat{\mathcal{D}} \log(m+n)$ maximizes, therefore giving the upper bound in Equation (22).

$\square$

**Lemma 17.** *The following condition is satisfied for Algorithm 1:*

$$\operatorname{tr}\left(\tilde{W}^t (H^t)^2\right) \leq \gamma h_\gamma(y_t, \bar{y}_t) + \gamma + 1. \tag{23}$$

*Proof.* The proof splits into two cases.

Case 1) $\gamma \leq y_t \bar{y}_t = y_t\left(\operatorname{tr}\left(\tilde{W}^t \tilde{X}^t\right) - 1\right)$:

Observe that $H^t = \mathbf{0}$ due to Lemma 12, giving $\operatorname{tr}\left(\tilde{W}^t (H^t)^2\right) = 0$. which demonstates (23) in this case.

Case 2) $\gamma > y_t \bar{y}_t = y_t\left(\operatorname{tr}\left(\tilde{W}^t \tilde{X}^t\right) - 1\right)$:
We have that
$$\operatorname{tr}\left(\tilde{W}^t (H^t)^2\right) = \operatorname{tr}\left(\tilde{W}^t (\tilde{X}^t)^2\right) \leq \operatorname{tr}\left(\tilde{W}^t \tilde{X}^t\right), \tag{24}$$

where the first equality comes from the fact $H^t = -y_t \tilde{X}^t$ from Lemma 12 and the second inequality comes from Lemma 14 and the fact that $(\tilde{X}^t)^2 \preceq \tilde{X}^t$ due to Lemma 9.

We split case 2 into two further subcases.

Sub-case 1) $\operatorname{tr}\left(\tilde{W}^t \tilde{X}^t\right) < \gamma + 1$ (Prediction smaller than margin):

Since we have
$$\operatorname{tr}\left(\tilde{W}^t \tilde{X}^t\right) < \gamma h_\gamma(y_t, \bar{y}_t) + \gamma + 1,$$

lower bounding the L.H.S. by (24) demonstrates (23).

Sub-case 2) $\operatorname{tr}\left(\tilde{W}^t \tilde{X}^t\right) \geq \gamma + 1$ (Prediction larger than margin with mistake):

We have
$$\operatorname{tr}\left(\tilde{W}^t \tilde{X}^t\right) \leq \left[\operatorname{tr}\left(\tilde{W}^t \tilde{X}^t\right) + \gamma - 1\right]_+ - (\gamma - 1) \leq \left[\operatorname{tr}\left(\tilde{W}^t \tilde{X}^t\right) + \gamma - 1\right]_+ + (\gamma + 1).$$

By the case 2 and sub-case 2 conditions we have that $y_t = -1$, with

$$\gamma h_\gamma(-1, \bar{y}_t) = \left[\gamma + \operatorname{tr}\left(\tilde{W}^t \tilde{X}^t\right) - 1\right]_+.$$

Thus we have

$$\mathrm{tr}\left(\tilde{\boldsymbol{W}}^t \tilde{\boldsymbol{X}}^t\right) \leq \gamma h_\gamma(-1, \bar{y}_t) + (\gamma + 1)$$

and by lower bounding L.H.S. by (24) we demonstrate (23) and thus the lemma. $\qquad\square$

Now we are ready to introduce the regret bound in terms of the hinge loss for the deterministic $\bar{y}_t$.

**Lemma 18.** *The hinge loss of Algorithm 1 with parameters* $\gamma \in (0, 1]$, $\widehat{\mathcal{D}} \geq \mathcal{D} \geq 1$, $\eta = \sqrt{\frac{\widehat{\mathcal{D}}\log(m+n)}{2T}}$, $T \geq 2\widehat{\mathcal{D}}\log(m + n)$ *and* $m + n \geq 3$, *is bounded by*

$$\sum_{t\in[T]} h_\gamma(y_t, \bar{y}_t) \leq \sum_{t\in[T]} h_\gamma(y_t, \bar{\boldsymbol{U}}_{i_t j_t}) + \frac{4}{\gamma}\sqrt{2\widehat{\mathcal{D}}\log(m+n)T} + \frac{4}{\gamma}\widehat{\mathcal{D}}\log(m+n), \quad (25)$$

*where* $\bar{\boldsymbol{U}}$, $\mathcal{D}$ *and their relationship are defined in the preliminaries of the proof.*

*Proof.* Substituting for $\bar{y}_t$ gives,

$$h_\gamma(y_t, \bar{y}_t) = \frac{1}{\gamma}[\gamma - y_t(\mathrm{tr}\left(\tilde{\boldsymbol{W}}^t \tilde{\boldsymbol{X}}^t\right) - 1)]_+ .$$

Lemma 7 gives,

$$h_\gamma(y_t, \bar{\boldsymbol{U}}_{i_t,j_t}) = \frac{1}{\gamma}[\gamma - y_t(\mathrm{tr}\left(\tilde{\boldsymbol{U}} \tilde{\boldsymbol{X}}^t\right) - 1)]_+ .$$

Define

$$f_t(\boldsymbol{Z}) := \frac{1}{\gamma}[\gamma - y_t(\mathrm{tr}\left(\boldsymbol{Z} \tilde{\boldsymbol{X}}^t\right) - 1)]_+$$

Since $h_\gamma(y_t, \cdot)$ is convex and the fact that a convex function applied to a linear function is again convex we have that $f(\cdot)$ is convex. We have

$$\sum_{t=1}^{T} \left(h_\gamma(y_t, \bar{y}_t) - h_\gamma(y_t, \bar{\boldsymbol{U}}_{i_t,j_t})\right) = \sum_{t=1}^{T} \left(f_t(\tilde{\boldsymbol{W}}^t) - f_t(\tilde{\boldsymbol{U}})\right)$$

$$\leq \sum_{t=1}^{T} \mathrm{tr}\left(\left(\tilde{\boldsymbol{W}}^t - \tilde{\boldsymbol{U}}\right)^\top \nabla f_t(\tilde{\boldsymbol{W}})\right) \quad (26)$$

$$= \sum_{t=1}^{T} \mathrm{tr}\left(\left(\tilde{\boldsymbol{W}}^t - \tilde{\boldsymbol{U}}\right)^\top \nabla_{\tilde{\boldsymbol{W}}} h_\gamma(y_t, \bar{y}_t)\right)$$

$$= \frac{1}{\gamma}\sum_{t=1}^{T} \mathrm{tr}\left((\tilde{\boldsymbol{W}}^t - \tilde{\boldsymbol{U}})\boldsymbol{H}^t\right), \quad (27)$$

where (26) follows from the fact that $f(\boldsymbol{A}) - f(\boldsymbol{B}) \leq \mathrm{tr}((\boldsymbol{A} - \boldsymbol{B})^\top \nabla f(\boldsymbol{A}))$ for a convex function $f$ and (27) comes from the definition of $\boldsymbol{H}^t = \nabla_{\tilde{\boldsymbol{W}}} \gamma h_\gamma(y_t, \bar{y}_t)$ (see (15)) and the fact that $\tilde{\boldsymbol{W}}^t$ and $\tilde{\boldsymbol{U}}$ are symmetric.

Therefore, we only need an upper bound to $\sum_{t=1}^{T} \mathrm{tr}\left((\tilde{\boldsymbol{W}}^t - \tilde{\boldsymbol{U}})\boldsymbol{H}^t\right)$. We observe that $\eta \in \left(0, \frac{1}{2}\right]$ due to the definition of $\eta$ and the assumption on $T$. Thus we can apply (22) and (23) to obtain

$$\sum_{t=1}^{T} \mathrm{tr}\left((\tilde{\boldsymbol{W}}^t - \tilde{\boldsymbol{U}})\boldsymbol{H}^t\right) \leq \frac{1}{\eta}\widehat{\mathcal{D}}\log(m+n) + \eta\gamma\sum_{t=1}^{T} h_\gamma(y_t, \bar{y}_t) + \eta(1+\gamma)T .$$

Substituing the above into (27) gives,

$$\sum_{t=1}^{T} h_\gamma(y_t, \bar{y}_t) \leq \sum_{t=1}^{T} h_\gamma(y_t, \bar{\boldsymbol{U}}_{i_t,j_t}) + \frac{1}{\gamma}\left(\frac{1}{\eta}\widehat{\mathcal{D}}\log(m+n) + \eta\gamma\sum_{t=1}^{T} h_\gamma(y_t, \bar{y}_t) + \eta(1+\gamma)T\right)$$

$$= \left(\frac{1}{1-\eta}\right)\left(\sum_{t=1}^{T} h_\gamma(y_t, \bar{\boldsymbol{U}}_{i_t,j_t}) + \frac{1}{\eta\gamma}\widehat{\mathcal{D}}\log(m+n) + \frac{\eta}{\gamma}(1+\gamma)T\right)$$

$$\leq \left(\frac{1}{1-\eta}\right)\left(\sum_{t=1}^{T} h_\gamma(y_t, \bar{U}_{i_t,j_t}) + \frac{1}{\eta\gamma}\widehat{\mathcal{D}}\log(m+n) + \frac{2\eta}{\gamma}T\right),$$

where the final inequality follows since $\gamma \in (0,1]$.

We apply $(1/(1-x)) \leq 1 + 2x$ for $x \in [0, 1/2]$ to obtain

$$\sum_{t=1}^{T} h_\gamma(y_t, \bar{y}_t) \leq (1+2\eta)\left(\sum_{t=1}^{T} h_\gamma(y_t, \bar{U}_{i_t,j_t}) + \frac{1}{\eta\gamma}\left(\widehat{\mathcal{D}}\log(m+n)\right) + \frac{2\eta}{\gamma}T\right)$$

$$= \underbrace{(1+2\eta)\sum_{t=1}^{T} h_\gamma(y_t, \bar{U}_{i_t,j_t})}_{(1)} + \underbrace{\frac{1}{\eta\gamma}\widehat{\mathcal{D}}\log(m+n)}_{(2)} + \underbrace{\frac{2\eta}{\gamma}T}_{(3)} + \underbrace{\frac{2}{\gamma}\widehat{\mathcal{D}}\log(m+n)}_{(4)} + \underbrace{\frac{4\eta^2}{\gamma}T}_{(5)}$$

Then, substituting the value for $\eta$,

$$\sum_{t=1}^{T} h_\gamma(y_t, \bar{y}_t) \leq \overbrace{\sum_{t=1}^{T} h_\gamma(y_t, \bar{U}_{i_t,j_t})}^{(a)} + \overbrace{\sqrt{\frac{2\widehat{\mathcal{D}}\log(m+n)}{T}}\sum_{t=1}^{T} h_\gamma(y_t, \bar{U}_{i_t,j_t})}^{(b)} +$$

$$\overbrace{\frac{4}{\gamma}\widehat{\mathcal{D}}\log(m+n)}^{(c)} + \overbrace{\frac{2}{\gamma}\sqrt{2\widehat{\mathcal{D}}\log(m+n)T}}^{(d)}$$

where $(1) = (a) + (b)$, $(2) + (3) = (d)$, and $(4) + (5) = (c)$. Recalling $\bar{U}_{i_t,j_t} = \gamma\tilde{U}_{i_t,j_t}$ from the preliminaries of the proof, we then have $h_\gamma(y_t, \bar{U}_{i_t,j_t}) \leq 2 \leq \frac{2}{\gamma}$, giving

$$\sum_{t=1}^{T} h_\gamma(y_t, \bar{y}_t) - \sum_{t=1}^{T} h_\gamma(y_t, \bar{U}_{i_t,j_t}) \leq \frac{4}{\gamma}\sqrt{2\widehat{\mathcal{D}}\log(m+n)T} + \frac{4}{\gamma}\widehat{\mathcal{D}}\log(m+n).$$

$$\square$$

Before we can compute the desired regret bound in terms of the 0-1 loss, we first need to introduce the following relationships.

**Lemma 19.** *For* $y_t \in \{-1, 1\}$, $\bar{y}_t \in \Re$, $Y_t \sim \text{UNIFORM}(-\gamma, \gamma)$, $\gamma \in (0, 1]$ *and* $\hat{y}_t := \text{sign}(\bar{y}_t - Y_t)$,

$$2\mathbb{E}[y_t \neq \hat{y}_t] \leq h_\gamma(y_t, \bar{y}_t).$$

*Proof.* We have

$$p(\hat{y}_t = 1) = \begin{cases} 0 & \text{if } \bar{y}_t \leq -\gamma \\ \frac{1}{2} + \frac{\bar{y}_t}{2\gamma} & \text{if } -\gamma < \bar{y}_t \leq \gamma \\ 1 & \text{if } \bar{y}_t > \gamma \end{cases}$$

and

$$p(\hat{y}_t = -1) = \begin{cases} 1 & \text{if } \bar{y}_t \leq -\gamma \\ \frac{1}{2} - \frac{\bar{y}_t}{2\gamma} & \text{if } -\gamma < \bar{y}_t \leq \gamma \\ 0 & \text{if } \bar{y}_t > \gamma. \end{cases}$$

The possible cases are as follows.

1. If $|\bar{y}_t| < \gamma$, $2\mathbb{E}[y_t \neq \hat{y}_t] = h_\gamma(y_t, \bar{y}_t)$. This is since if $y_t = 1$, $\mathbb{E}[y_t \neq \hat{y}_t] = \frac{1}{2} - \frac{\bar{y}_t}{2\gamma}$ and $h_\gamma(y_t, \bar{y}_t) = \frac{1}{\gamma}(\gamma - \bar{y}_t)$. Similarly if $y_t = -1$, $\mathbb{E}[y_t \neq \hat{y}_t] = \frac{1}{2} + \frac{\bar{y}_t}{2\gamma}$ and $h_\gamma(y_t, \bar{y}_t) = \frac{1}{\gamma}(\gamma + \bar{y}_t)$.

2. If $|\bar{y}_t| \geq \gamma$ and $\mathbb{E}[y_t \neq \hat{y}_t] = 0$, then $h_\gamma(y_t, \bar{y}_t) = \frac{1}{\gamma}[\gamma - |\bar{y}_t|]_+ = 0$.

3. If $|\bar{y}_t| \geq \gamma$ and $\mathbb{E}[y_t \neq \hat{y}_t] = 1$, $h_\gamma(y_t, \bar{y}_t) = \frac{1}{\gamma}[\gamma + |\bar{y}_t|]_+ \geq \frac{2\gamma}{\gamma} = 2\mathbb{E}[y_t \neq \hat{y}_t]$.

$\square$

**Lemma 20.** *Suppose we have $\boldsymbol{U}$ as in Theorem 1. Recalling that $\bar{\boldsymbol{U}} = \gamma\boldsymbol{U}$ from the preliminaries of the proof, we have that*

$$h_\gamma(y_t, \bar{\boldsymbol{U}}_{i_t,j_t}) \leq 2[y_t \neq U_{i_t j_t}].$$

*Proof.* Recall that $\boldsymbol{U} \in \{-1,1\}^{m \times n}$. The hinge loss is then given by $h_\gamma(y_t, \bar{\boldsymbol{U}}_{i_t,j_t}) = \frac{1}{\gamma}[\gamma - y_t\bar{\boldsymbol{U}}_{i_t,j_t}]_+$. In the case that $y_t = \bar{\boldsymbol{U}}_{i_t,j_t}$, $h_\gamma(y_t, \bar{\boldsymbol{U}}_{i_t,j_t}) = \frac{1}{\gamma}[\gamma - \gamma]_+ = 0$. Otherwise, $h_\gamma(y_t, \bar{\boldsymbol{U}}_{i_t,j_t}) = \frac{1}{\gamma}[\gamma + \gamma]_+ = 2$. $\square$

We proceed by giving a sharper bound for Algorithm 1 than is stated in Theorem 1. This, however, only holds under the additional assumption that $T \geq 2\widehat{\mathcal{D}}\log(m+n)$.

**Theorem 21.** *The expected regret of Algorithm 1 with* **non-conservative** *updates and parameters $\gamma \in (0,1]$, $\widehat{\mathcal{D}} \geq \mathcal{D}^\gamma_{\boldsymbol{M},\boldsymbol{N}}(\boldsymbol{U})$, $\eta = \sqrt{\frac{\widehat{\mathcal{D}}\log(m+n)}{2T}}$, p.d. matrices $\boldsymbol{M} \in \boldsymbol{S}^m_{++}$ and $\boldsymbol{N} \in \boldsymbol{S}^n_{++}$ and for $T \geq 2\widehat{\mathcal{D}}\log(m+n)$ is bounded by*

$$\mathbb{E}[|\mathbb{M}|] - \sum_{t \in [T]} [y_t \neq U_{i_t j_t}] \leq \frac{2}{\gamma}\sqrt{2\widehat{\mathcal{D}}\log(m+n)T} + \frac{2}{\gamma}\widehat{\mathcal{D}}\log(m+n) \tag{28}$$

*for all $\boldsymbol{U} \in \{-1,1\}^{m \times n}$ with $\|\boldsymbol{U}\|_{max} \leq 1/\gamma$.*

*Proof.* Starting from Lemma 18, we observe that we can apply Lemma 19, to bound the expected mistakes of the latter by the cumulative hinge loss of the former. Combining this with Lemma 20 then gives the desired regret bound in terms of the 0-1 loss. Note that $\widehat{\mathcal{D}} \geq \mathcal{D} = \mathcal{D}^\gamma_{\boldsymbol{M},\boldsymbol{N}}(\boldsymbol{U})$, where the inequality comes from Lemma 18 and the equality is stated in the preliminaries of the proof, which assumes that $\|\boldsymbol{U}\|_{\max} \leq 1/\gamma$. $\square$

We will now prove the first part of Theorem 1. We split into two cases,

Case 1) $T < 2\widehat{\mathcal{D}}\log(m+n)$:

We have that

$$\mathbb{E}[|\mathbb{M}|] - \sum_{t \in [T]} [y_t \neq U_{i_t j_t}] \leq T < 2\widehat{\mathcal{D}}\log(m+n)$$

for all $\eta > 0$.

Case 2) $T \geq 2\widehat{\mathcal{D}}\log(m+n)$:

From (28) we have

$$\mathbb{E}[|\mathbb{M}|] - \sum_{t \in [T]} [y_t \neq U_{i_t j_t}] \leq \frac{2}{\gamma}\sqrt{2\widehat{\mathcal{D}}\log(m+n)T} + \frac{2}{\gamma}\widehat{\mathcal{D}}\log(m+n)$$

$$= \frac{2}{\gamma}\sqrt{2\widehat{\mathcal{D}}\log(m+n)T} + \frac{2}{\gamma}\sqrt{(\widehat{\mathcal{D}}\log(m+n))^2}$$

$$\leq \frac{2}{\gamma}\sqrt{2\widehat{\mathcal{D}}\log(m+n)T} + \frac{2}{\gamma}\sqrt{\frac{1}{2}\widehat{\mathcal{D}}\log(m+n)T} \tag{29}$$

$$= \frac{2}{\gamma}\left(\sqrt{2} + \sqrt{\frac{1}{2}}\right)\sqrt{\widehat{\mathcal{D}}\log(m+n)T}$$

where in Equation (29) we used the assumption on $T$.

Combining both cases, we have that the following holds for all $T$

$$\mathbb{E}[|\mathbb{M}|] - \sum_{t \in [T]} [y_t \neq U_{i_t j_t}] \leq \frac{2}{\gamma}\left(\sqrt{2} + \sqrt{\frac{1}{2}}\right)\sqrt{\widehat{\mathcal{D}}\log(m+n)T} + \min(2\widehat{\mathcal{D}}\log(m+n), T)$$

$$= \frac{2}{\gamma} \left( \sqrt{2} + \sqrt{\frac{1}{2}} \right) \sqrt{\widehat{\mathcal{D}} \log(m+n)T} + \sqrt{\min(2\widehat{\mathcal{D}} \log(m+n), T)^2}$$

$$\leq \frac{2}{\gamma} \left( \sqrt{2} + \sqrt{\frac{1}{2}} \right) \sqrt{\widehat{\mathcal{D}} \log(m+n)T} + \sqrt{2\widehat{\mathcal{D}} \log(m+n)T}$$

$$\leq \frac{2}{\gamma} \left( \sqrt{2} + \sqrt{\frac{1}{2}} \right) \sqrt{\widehat{\mathcal{D}} \log(m+n)T} + \frac{2}{\gamma} \sqrt{\frac{1}{2}\widehat{\mathcal{D}} \log(m+n)T} \quad (30)$$

$$= \frac{4}{\gamma} \sqrt{2\widehat{\mathcal{D}} \log(m+n)T}$$

where we used the fact that $\frac{1}{\gamma} \geq 1$ in Equation (30). Thus we have demonstrated (5) proving Theorem 1. ∎

### B.3 Proof for the Realizable Case

In this subsection, we prove the second part of Theorem 1. Recall from the theorem statement that $y_t = U_{i_t j_t}$ for all $t \in \mathbb{M}$, $\mathrm{mc}(U)^{-1} \geq \gamma$, $\widehat{\mathcal{D}} \geq \min_{V \in \mathrm{SP}^1(U)} \mathcal{D}_{M,N}^\gamma(V)$, $\eta = \gamma$ and that we have **conservative** updates. Recall from the preliminaries of the proof that given $U, \bar{U} \in \Re^{m \times n}$ is such that $\bar{U} = \gamma \operatorname*{argmin}_{V \in \mathrm{SP}^1(U)} \mathcal{D}_{M,N}^\gamma(V)$, meaning that $\min_{i \in [m], j \in [n]} |\bar{U}_{ij}| \geq \gamma$. Also recall that $\mathcal{D} = \min_{V \in \mathrm{SP}^1(U)} \mathcal{D}_{M,N}^\gamma(V)$.

**Lemma 22.** *[25, Lemma 2.1] If $A \in S_+^d$ with eigenvalues in $[0, 1]$ and $a \in \Re$ then:*

$$(1 - e^a) A \preceq I - \exp(aA)$$

**Lemma 23.** *For all trials $t \in \mathbb{M}$, we have:*

$$\Delta(\tilde{U}, \tilde{W}^t) - \Delta(\tilde{U}, \tilde{W}^{t+1}) \geq \eta y_t \operatorname{tr}\left(\tilde{U} \tilde{X}^t\right) + (1 - e^{\eta y_t}) \operatorname{tr}\left(\tilde{W}^t \tilde{X}^t\right) . \quad (31)$$

*Proof.* We have:

$$\Delta(\tilde{U}, \tilde{W}^t) - \Delta(\tilde{U}, \tilde{W}^{t+1}) = \operatorname{tr}\left(\tilde{U} \log \tilde{W}^{t+1} - \tilde{U} \log \tilde{W}^t\right) + \operatorname{tr}\left(\tilde{W}^t\right) - \operatorname{tr}\left(\tilde{W}^{t+1}\right)$$

$$= \eta y_t \operatorname{tr}\left(\tilde{U} \tilde{X}^t\right) + \operatorname{tr}\left(\tilde{W}^t\right) - \operatorname{tr}\left(e^{\log \tilde{W}^t + \eta y_t \tilde{X}^t}\right) \quad (32)$$

$$\geq \eta y_t \operatorname{tr}\left(\tilde{U} \tilde{X}^t\right) + \operatorname{tr}\left(\tilde{W}^t\right) - \operatorname{tr}\left(e^{\log \tilde{W}^t} e^{\eta y_t \tilde{X}^t}\right) \quad (33)$$

$$= \eta y_t \operatorname{tr}\left(\tilde{U} \tilde{X}^t\right) + \operatorname{tr}\left(\tilde{W}^t \left(I - e^{\eta y_t \tilde{X}^t}\right)\right)$$

$$\geq \eta y_t \operatorname{tr}\left(\tilde{U} \tilde{X}^t\right) + (1 - e^{\eta y_t}) \operatorname{tr}\left(\tilde{W}^t \tilde{X}^t\right), \quad (34)$$

where Equation (32) comes from the update of the algorithm, Equation (33) comes from Lemma 11 and Equation (34) comes from Lemma 22 which applies since, by Lemma 9 all eigenvalues of $\tilde{X}^t$ are in $[0, 1]$. □

**Lemma 24.** *[24, Lemma A.5] For $x \in [-1, 1]$,*

$$x^2 + x + 1 - e^x \geq (3 - e)x^2 .$$

We proceed by showing that the "progress" $\Delta(\tilde{U}, \tilde{W}^t) - \Delta(\tilde{U}, \tilde{W}^{t+1})$ of $\tilde{W}^t$ towards $\tilde{U}$ may be further lower bounded by $c\gamma$ (see Lemma 25).

**Lemma 25.** *Let $c := 3 - e$. For all trials $t$ with $t \in \mathbb{M}$ (under the conditions of Lemma 28) we have:*

$$\Delta(\tilde{U}, \tilde{W}^t) - \Delta(\tilde{U}, \tilde{W}^{t+1}) \geq c\gamma^2$$

*Proof.* By Lemma 7, $\bar{U}_{i_t j_t} = \text{tr}\left(\tilde{U}\tilde{X}^t\right) - 1$ so since $y_t = \text{sign}(\bar{U}_{i_t j_t})$, and $\gamma \leq |\bar{U}_{i_t j_t}|$ we have $\gamma \leq y_t\left(\text{tr}\left(\tilde{U}\tilde{X}^t\right) - 1\right)$. So when $y_t = 1$ we have $\text{tr}\left(\tilde{U}\tilde{X}^t\right) \geq 1 + \gamma$ and when $y_t = -1$ we have $\text{tr}\left(\tilde{U}\tilde{X}^t\right) \leq 1 - \gamma$. We use these inequalities as follows.

First suppose that $y_t = 1$. By Lemma 23 we have:

$$\Delta(\tilde{U}, \tilde{W}^t) - \Delta(\tilde{U}, \tilde{W}^{t+1}) \geq \gamma \,\text{tr}\left(\tilde{U}\tilde{X}^t\right) + (1 - e^\gamma)\,\text{tr}\left(\tilde{W}^t\tilde{X}^t\right)$$
$$\geq \gamma\,(1 + \gamma) + (1 - e^\gamma)\,\text{tr}\left(\tilde{W}^t\tilde{X}^t\right)$$
$$\geq \gamma\,(1 + \gamma) + (1 - e^\gamma) \tag{35}$$
$$= (\gamma + \gamma^2) + 1 - e^\gamma$$
$$\geq c\gamma^2, \tag{36}$$

where Equation (36) comes from Lemma 24 and Equation (35) comes from the fact that $\hat{y}^t = -1$ and hence, by the algorithm, $\text{tr}\left(\tilde{W}^t\tilde{X}^t\right) \leq 1$.

Now suppose that $y_t = -1$. By Lemma 23 we have:

$$\Delta(\tilde{U}, \tilde{W}^t) - \Delta(\tilde{U}, \tilde{W}^{t+1}) \geq -\gamma\,\text{tr}\left(\tilde{U}\tilde{X}^t\right) + \left(1 - e^{-\gamma}\right)\text{tr}\left(\tilde{W}^t\tilde{X}^t\right)$$
$$\geq -\gamma\,(1 - \gamma) + \left(1 - e^{-\gamma}\right)\text{tr}\left(\tilde{W}^t\tilde{X}^t\right)$$
$$\geq -\gamma\,(1 - \gamma) + \left(1 - e^{-\gamma}\right) \tag{37}$$
$$= -\gamma + \gamma^2 + 1 - e^{-\gamma}$$
$$\geq c\gamma^2, \tag{38}$$

where Equation (38) comes from Lemma 24 and Equation (37) comes from the fact that $\hat{y}^t = 1$ and hence, by the algorithm, $\text{tr}\left(\tilde{W}^t\tilde{X}^t\right) \geq 1$. $\square$

**Lemma 26.** *We have,*
$$c\gamma^2|\mathbb{M}| \leq \Delta(\tilde{U}, \tilde{W}^1)\,.$$

*Proof.* Suppose that we have $T$ trials. Then we have:

$$\Delta(\tilde{U}, \tilde{W}^1) \geq \Delta(\tilde{U}, \tilde{W}^1) - \Delta(\tilde{U}, \tilde{W}^{T+1})$$
$$= \sum_{t \in [T]} \left(\Delta(\tilde{U}, \tilde{W}^t) - \Delta(\tilde{U}, \tilde{W}^{t+1})\right)$$
$$= \sum_{t \in \mathbb{M}} \left(\Delta(\tilde{U}, \tilde{W}^t) - \Delta(\tilde{U}, \tilde{W}^{t+1})\right) \tag{39}$$
$$\geq \sum_{t \in \mathbb{M}} c\gamma^2 \tag{40}$$
$$= c\gamma^2|\mathbb{M}|,$$

where (40) follows from (39) using Lemma 25. $\square$

**Lemma 27.** *Given that $\tilde{W}^1 = \widehat{\mathcal{D}}\frac{I}{m+n}$ we have*

$$\Delta(\tilde{U}, \tilde{W}^1) \leq \text{tr}\left(\tilde{U}\right)\log(m+n) + \text{tr}(\tilde{U})\log\frac{\text{tr}(\tilde{U})}{\widehat{\mathcal{D}}} + \widehat{\mathcal{D}} - \text{tr}(\tilde{U})$$

*Proof.* We have:

$$\Delta(\tilde{U}, \tilde{W}^1) = \text{tr}\left(\tilde{U}\log\tilde{U}\right) - \text{tr}\left(\tilde{U}\log\tilde{W}^1\right) + \text{tr}\left(\tilde{W}^1\right) - \text{tr}(\tilde{U})$$

$$= \operatorname{tr}\left(\tilde{\boldsymbol{U}} \log \tilde{\boldsymbol{U}}\right) - \operatorname{tr}\left(\tilde{\boldsymbol{U}} \log\left(\frac{\widehat{\mathcal{D}}}{m+n}\boldsymbol{I}\right)\right) + \operatorname{tr}\left(\frac{\widehat{\mathcal{D}}}{m+n}\boldsymbol{I}\right) - \operatorname{tr}(\tilde{\boldsymbol{U}})$$

$$= \operatorname{tr}\left(\tilde{\boldsymbol{U}} \log \tilde{\boldsymbol{U}}\right) - \operatorname{tr}\left(\tilde{\boldsymbol{U}} \log\left(\frac{\widehat{\mathcal{D}}}{m+n}\boldsymbol{I}\right)\right) + \widehat{\mathcal{D}} - \operatorname{tr}(\tilde{\boldsymbol{U}})$$

$$= \operatorname{tr}\left(\tilde{\boldsymbol{U}} \log \tilde{\boldsymbol{U}}\right) - \operatorname{tr}\left(\tilde{\boldsymbol{U}}\left(\boldsymbol{I} \log\left(\frac{\widehat{\mathcal{D}}}{m+n}\right)\right)\right) + \widehat{\mathcal{D}} - \operatorname{tr}(\tilde{\boldsymbol{U}})$$

$$= \operatorname{tr}\left(\tilde{\boldsymbol{U}} \log \tilde{\boldsymbol{U}}\right) - \operatorname{tr}\left(\tilde{\boldsymbol{U}} \log\left(\frac{\widehat{\mathcal{D}}}{m+n}\right)\right) + \widehat{\mathcal{D}} - \operatorname{tr}(\tilde{\boldsymbol{U}}) \tag{41}$$

$$\leq \operatorname{tr}\left(\tilde{\boldsymbol{U}} \log(\operatorname{tr}(\tilde{\boldsymbol{U}}))\right) - \operatorname{tr}\left(\tilde{\boldsymbol{U}} \log\left(\frac{\widehat{\mathcal{D}}}{m+n}\right)\right) + \widehat{\mathcal{D}} - \operatorname{tr}(\tilde{\boldsymbol{U}}) \tag{42}$$

$$= \left(\log(\operatorname{tr}(\tilde{\boldsymbol{U}})) - \log\left(\frac{\widehat{\mathcal{D}}}{m+n}\right)\right)\operatorname{tr}(\tilde{\boldsymbol{U}}) + \widehat{\mathcal{D}} - \operatorname{tr}(\tilde{\boldsymbol{U}})$$

$$= \log\left(\frac{\operatorname{tr}(\tilde{\boldsymbol{U}})(m+n)}{\widehat{\mathcal{D}}}\right)\operatorname{tr}(\tilde{\boldsymbol{U}}) + \widehat{\mathcal{D}} - \operatorname{tr}(\tilde{\boldsymbol{U}}),$$

where (42) follows from (41), since $\tilde{\boldsymbol{U}} := \boldsymbol{V}\boldsymbol{\Lambda}\boldsymbol{V}^{-1}$ where $\boldsymbol{\Lambda}$ is a diagonal matrix of the eigenvalues of $\tilde{\boldsymbol{U}}$. This holds since,

$$\begin{aligned}
\operatorname{tr}(\tilde{\boldsymbol{U}} \log \tilde{\boldsymbol{U}}) &= \operatorname{tr}(\boldsymbol{V}\boldsymbol{\Lambda}\boldsymbol{V}^{-1}\boldsymbol{V} \log \boldsymbol{\Lambda}\boldsymbol{V}^{-1}) \\
&= \operatorname{tr}(\boldsymbol{V}\boldsymbol{\Lambda} \log \boldsymbol{\Lambda}\boldsymbol{V}^{-1}) \\
&= \operatorname{tr}(\boldsymbol{\Lambda} \log \boldsymbol{\Lambda}) \\
&= \sum_{i=1}^{m+n} \lambda_i \log(\lambda_i) \\
&\leq (\sum_{i=1}^{m+n} \lambda_i) \log(\sum_{i=1}^{m+n} \lambda_i) \\
&= \operatorname{tr}(\tilde{\boldsymbol{U}} \log(\operatorname{tr}(\tilde{\boldsymbol{U}}))).
\end{aligned}$$

$\square$

**Lemma 28.** *The mistakes, $|\mathbb{M}|$, of Algorithm 1 with the assumption that $y_t = \operatorname{sign}(\bar{U}_{i_t j_t})$ for all $t \in \mathbb{M}$ and with parameters $\gamma \leq \operatorname{mc}(\boldsymbol{U})^{-1}$, $1 \leq \widehat{\mathcal{D}}$ and $\eta = \gamma$ and* **conservative** *updates, is bounded above by:*

$$|\mathbb{M}| \leq 3.6\frac{1}{\gamma^2}\left(\mathcal{D}\left(\log(m+n) + \log\frac{\mathcal{D}}{\widehat{\mathcal{D}}}\right) + \widehat{\mathcal{D}} - \mathcal{D}\right) \tag{43}$$

*Proof.* Combining Lemmas 26 and 27 gives us

$$|\mathbb{M}| \leq \frac{1}{c}\frac{1}{\gamma^2}\left(\operatorname{tr}\left(\tilde{\boldsymbol{U}}\right)\log(m+n) + \operatorname{tr}(\tilde{\boldsymbol{U}})\log\frac{\operatorname{tr}(\tilde{\boldsymbol{U}})}{\widehat{\mathcal{D}}} + \widehat{\mathcal{D}} - \operatorname{tr}(\tilde{\boldsymbol{U}})\right)$$

Using Lemma 8 and upper bounding $1/c$ by 3.6 then gives the result. $\square$

The theorem statement for the realizable case then follows by setting $\widehat{\mathcal{D}} \geq \mathcal{D}$. $\blacksquare$

## C Proof of Theorem 3

We recall Theorem 3 and then prove it.

**Theorem 3.** *If $U \in \mathbb{B}_{k,\ell}^{m,n}$ define*

$$\mathcal{D}_{M,N}^{\circ}(U) := \begin{cases} 2\operatorname{tr}(R^{\top}MR)\mathcal{R}_M + 2\operatorname{tr}(C^{\top}NC)\mathcal{R}_N + 2k + 2\ell & M \text{ and } N \text{ are PDLaplacians} \\ k\operatorname{tr}(R^{\top}MR)\mathcal{R}_M + \ell\operatorname{tr}(C^{\top}NC)\mathcal{R}_N & M \in S_{++}^m \text{ and } N \in S_{++}^n \end{cases}.$$

*as the minimum over all decompositions of $U = RU^*C^{\top}$ for $R \in \mathcal{B}^{m,k}$, $C \in \mathcal{B}^{n,\ell}$ and $U^* \in \{-1,1\}^{k\times\ell}$. Thus for $U \in \mathbb{B}_{k,\ell}^{m,n}$,*

$$\mathcal{D}_{M,N}^{\gamma}(U) \leq \mathcal{D}_{M,N}^{\circ}(U) \qquad (\text{if } \|U\|_{max} \leq 1/\gamma)$$

$$\min_{V \in \mathrm{SP}^1(U)} \mathcal{D}_{M,N}^{\gamma}(V) \leq \mathcal{D}_{M,N}^{\circ}(U) \qquad (\text{if } \mathrm{mc}(U) \leq 1/\gamma).$$

*Proof.* A $\gamma$-*decomposition* of matrix $U$ is given by a $\hat{P} \in \mathcal{N}^{m,d}$ and a $\hat{Q} \in \mathcal{N}^{n,d}$ such that $\hat{P}\hat{Q}^{\top} = \gamma U$. A *block-invariant decomposition* of matrix $U \in \mathbb{B}_{k,\ell}^{m,n}$ is given by a $\hat{P} \in \mathcal{N}^{m,d}$ and a $\hat{Q} \in \mathcal{N}^{n,d}$ for some $d$ such that there exists a $\delta \in (0,1]$, $\hat{P}^* \in \mathcal{N}^{k,d}$, and a $\hat{Q}^* \in \mathcal{N}^{\ell,d}$, so that $\hat{P} = R\hat{P}^*$, $\hat{Q} = C\hat{Q}^*$ and $\hat{P}\hat{Q}^{\top} = \delta U$.

We now prove the following intermediate result,

> **Lemma:** If $U \in \mathbb{B}_{k,\ell}^{m,n}$, then for every $\gamma \in (0, 1/\|U\|_{\max})$, there exists a block-invariant $\gamma$-decomposition of $U$.

> *Proof.* Since $U \in \mathbb{B}_{k,\ell}^{m,n}$ we have that $U = RU^*C^{\top}$ for some $R \in \mathcal{B}^{m,k}$, $C \in \mathcal{B}^{n,\ell}$ and $U^* \in \{-1,1\}^{k\times\ell}$. Observe by block invariance we have that $\|U\|_{\max} = \|U^*\|_{\max}$ and by the definition of $\|\cdot\|_{\max}$ we have that there exists a $\left(\frac{1}{\|U\|_{\max}}\right)$-decomposition of $U^*$ via factors $\hat{P}^* \in \mathcal{N}^{k,d}$, and a $\hat{Q}^* \in \mathcal{N}^{\ell,d}$, this implies that $\hat{P} := R\hat{P}^*$, $\hat{Q} := C\hat{Q}^*$ is a $\left(\frac{1}{\|U\|_{\max}}\right)$-block-invariant decomposition of $U$.

> Now given any $\gamma \in (0, 1/\|U\|_{\max})$ we construct a $\gamma$-block-invariant decomposition of $U$. Set $c := \gamma\|U\|_{\max}$. We construct new factor matrices $\hat{P}' \in \mathcal{N}^{m,d+1}$ and $\hat{Q}' \in \mathcal{N}^{n,d+1}$

> $$\hat{P}' := \begin{pmatrix} c\hat{P} & (\sqrt{1-c^2})\mathbf{1} \end{pmatrix}; \qquad \hat{Q}' := \begin{pmatrix} \hat{Q} & 0 \end{pmatrix}.$$

> Observe that $(\hat{P}', \hat{Q}')$ is the required $\gamma$-block-invariant decomposition of $U$. $\quad\square$

Recall (3),

$$\mathcal{D}_{M,N}^{\gamma}(U) := \min_{\hat{P}\hat{Q}^{\top}=\gamma U} \mathcal{R}_M \operatorname{tr}\left(\hat{P}^{\top}M\hat{P}\right) + \mathcal{R}_N \operatorname{tr}\left(\hat{Q}^{\top}N\hat{Q}\right). \tag{44}$$

Observe that when the feasible set of the optimization that defines $\mathcal{D}_{M,N}^{\gamma}(U)$ is non-empty and $U \in \mathbb{B}_{k,\ell}^{m,n}$, there exists a member of the feasible set which is a block-invariant decomposition of $U$ by the lemma above. We proceed by proving an upper bound of

$$\mathcal{R}_M \operatorname{tr}\left(\hat{P}^{\top}M\hat{P}\right) + \mathcal{R}_N \operatorname{tr}\left(\hat{Q}^{\top}N\hat{Q}\right)$$

for every block-invariant decomposition of $U$.

First we will bound the term $\operatorname{tr}(\hat{P}^{\top}M\hat{P})$ for general positive definite matrices and then for PD-Laplacians. By symmetry, the bound will also hold for $\operatorname{tr}(\hat{Q}^{\top}N\hat{Q})$.

Suppose $(\hat{P}, \hat{Q})$ is a block-invariant decomposition of $U$. Then, we have

$$\operatorname{tr}(\hat{P}^{\top}M\hat{P}) = \operatorname{tr}((R\hat{P}^*)^{\top}MR\hat{P}^*) = \operatorname{tr}(\hat{P}^*(\hat{P}^*)^{\top}R^{\top}MR)$$
$$\leq \operatorname{tr}(\hat{P}^*(\hat{P}^*)^{\top})\operatorname{tr}(R^{\top}MR) = k\operatorname{tr}(R^{\top}MR),$$

where the inequality comes from the fact that $\mathrm{tr}(\boldsymbol{AB}) \le \lambda_{\max}(\boldsymbol{A})\,\mathrm{tr}(\boldsymbol{B}) \le \mathrm{tr}(\boldsymbol{A})\,\mathrm{tr}(\boldsymbol{B})$ for $\boldsymbol{A}, \boldsymbol{B} \in \boldsymbol{S}_+$. By symmetry we have demonstrated the inequality for positive definite matrices.

We now consider PDLaplacians. Assume $\boldsymbol{M} := \boldsymbol{L}^\circ = \boldsymbol{L} + \left(\frac{1}{m}\right)\left(\frac{1}{m}\right)^\top \mathcal{R}_{\boldsymbol{L}}^{-1}$, a PDLaplacian. Recall the following two elementary inequalities from the preliminaries: if $\boldsymbol{u} \in [-1,1]^m$, then

$$(\boldsymbol{u}^\top \boldsymbol{L} \boldsymbol{u})\mathcal{R}_{\boldsymbol{L}} \le \frac{1}{2}(\boldsymbol{u}^\top \boldsymbol{L}^\circ \boldsymbol{u})\mathcal{R}_{\boldsymbol{L}^\circ}\,, \tag{45}$$

$$(\boldsymbol{u}^\top \boldsymbol{L}^\circ \boldsymbol{u})\mathcal{R}_{\boldsymbol{L}^\circ} \le 2(\boldsymbol{u}^\top \boldsymbol{L} \boldsymbol{u}\,\mathcal{R}_{\boldsymbol{L}} + 1)\,. \tag{46}$$

Observe that for an $m \times m$ graph Laplacian $\boldsymbol{L}$ with adjacency matrix $\boldsymbol{A}$ that for $\boldsymbol{X} \in \Re^{m \times d}$,

$$\mathrm{tr}(\boldsymbol{X}^\top \boldsymbol{L} \boldsymbol{X}) = \sum_{(i,j)\in E} A_{ij} \left\| \boldsymbol{X}_i - \boldsymbol{X}_j \right\|^2\,. \tag{47}$$

Suppose $(\hat{\boldsymbol{P}}, \hat{\boldsymbol{Q}})$ is a block-invariant decomposition of $\boldsymbol{U}$ then the row vectors $\hat{\boldsymbol{P}}_1, \ldots, \hat{\boldsymbol{P}}_m$ come in at most $k$ distinct varieties, that is $|\bigcup_{i \in [m]} \hat{\boldsymbol{P}}_i| \le k$. The same holds for $\boldsymbol{R}$ and furthermore $(\hat{\boldsymbol{P}}_r = \hat{\boldsymbol{P}}_s) \iff (\boldsymbol{R}_r = \boldsymbol{R}_s)$ for $r, s \in [m]$. Observe that given $r, s \in [m]$ that if $\boldsymbol{R}_r \ne \boldsymbol{R}_s$ then $\|\boldsymbol{R}_r - \boldsymbol{R}_s\|^2 = 2$ and $\left\|\hat{\boldsymbol{P}}_r - \hat{\boldsymbol{P}}_s\right\|^2 \le 4$ since they are coordinate and unit vectors respectively. This then implies,

$$\mathrm{tr}\left(\hat{\boldsymbol{P}}^\top \boldsymbol{L} \hat{\boldsymbol{P}}\right) \le 2\,\mathrm{tr}\left(\boldsymbol{R}^\top \boldsymbol{L} \boldsymbol{R}\right)\,. \tag{48}$$

Thus we have

$$
\begin{aligned}
\mathrm{tr}(\hat{\boldsymbol{P}}^\top \boldsymbol{M} \hat{\boldsymbol{P}})\mathcal{R}_{\boldsymbol{M}} &\le 2\,\mathrm{tr}(\hat{\boldsymbol{P}}^\top \boldsymbol{L} \hat{\boldsymbol{P}})\mathcal{R}_{\boldsymbol{L}} + 2k && \text{by (46)}\\
&\le 4\,\mathrm{tr}\left(\boldsymbol{R}^\top \boldsymbol{L} \boldsymbol{R}\right)\mathcal{R}_{\boldsymbol{L}} + 2k && \text{by (48)}\\
&\le 2\,\mathrm{tr}\left(\boldsymbol{R}^\top \boldsymbol{M} \boldsymbol{R}\right)\mathcal{R}_{\boldsymbol{M}} + 2k && \text{by (45)}
\end{aligned}
$$

By symmetry we have demonstrated the inequality for PDLaplacians. $\qquad \square$

## D Proofs for Section 5

### D.1 Proof of Proposition 4

We now prove Proposition 4, that the transductive and inductive algorithms are equivalent. Recall by assumption that $\mathcal{R}_{\mathcal{M}} = \mathcal{R}_{\boldsymbol{M}}$ and $\mathcal{R}_{\mathcal{N}} = \mathcal{R}_{\boldsymbol{N}}$.

#### D.1.1 Equivalence of Traces

Suppose, in this subsection, that we have some given trial $t$. In this subsection we analyse the inductive algorithm. We make the following definitions:

**Definition 29.** *For all* $s \in \mathbb{U} \cap [t]$

$$\boldsymbol{v}(s) := \left[ \frac{(\sqrt{(\boldsymbol{M}^t)^+})\boldsymbol{e}^{i_s}}{\sqrt{2\mathcal{R}_{\boldsymbol{M}}}}; \frac{(\sqrt{(\boldsymbol{N}^t)^+})\boldsymbol{e}^{j_s}}{\sqrt{2\mathcal{R}_{\boldsymbol{N}}}} \right]$$

$$\bar{\boldsymbol{v}}(s) := \left[ \frac{(\sqrt{(\boldsymbol{M}^{T+1})^+})\boldsymbol{e}^{i_s}}{\sqrt{2\mathcal{R}_{\boldsymbol{M}}}}; \frac{(\sqrt{(\boldsymbol{N}^{T+1})^+})\boldsymbol{e}^{j_s}}{\sqrt{2\mathcal{R}_{\boldsymbol{N}}}} \right]$$

Note that $\tilde{\boldsymbol{X}}^t(s) = \boldsymbol{v}(s)\boldsymbol{v}(s)^\top$ and $\tilde{\boldsymbol{X}}^{T+1}(s) = \bar{\boldsymbol{v}}(s)\bar{\boldsymbol{v}}(s)^\top$ for $s \in \mathbb{U} \cap [t]$.

**Lemma 30.** *For all* $l \in \mathbb{N}$ *and for all* $a_1, a_2, ..., a_l \in \mathbb{U} \cap [t-1]$ *there exists some* $\alpha \in \Re$ *such that:*

$$\tilde{\boldsymbol{X}}^t(a_1)\tilde{\boldsymbol{X}}^t(a_2)\cdots\tilde{\boldsymbol{X}}^t(a_l) = \alpha \boldsymbol{v}(a_1)\boldsymbol{v}(a_l)^\top$$

*and*

$$\tilde{\boldsymbol{X}}^{T+1}(a_1)\tilde{\boldsymbol{X}}^{T+1}(a_2)\cdots\tilde{\boldsymbol{X}}^{T+1}(a_l) = \alpha \bar{\boldsymbol{v}}(a_1)\bar{\boldsymbol{v}}(a_l)^\top$$

*Proof.* We prove by induction on $l$. In the case $l := 1$ the result is clear with $\alpha := 1$.

Now suppose the result holds with $l := q$ for some $q \in \mathbb{N}$. We now show that it holds for $l := q + 1$. Since it holds for $l := q$, choose $\alpha'$ such that $\tilde{\boldsymbol{X}}^t(a_1)\tilde{\boldsymbol{X}}^t(a_2)\cdots\tilde{\boldsymbol{X}}^t(a_q) = \alpha'\boldsymbol{v}(a_1)\boldsymbol{v}(a_q)^\top$ and $\tilde{\boldsymbol{X}}^{T+1}(a_1)\tilde{\boldsymbol{X}}^{T+1}(a_2)\cdots\tilde{\boldsymbol{X}}^{T+1}(a_q) = \alpha'\bar{\boldsymbol{v}}(a_1)\bar{\boldsymbol{v}}(a_q)^\top$. Note that we now have:

$$
\begin{aligned}
\tilde{\boldsymbol{X}}^t(a_1)\tilde{\boldsymbol{X}}^t(a_2)\cdots\tilde{\boldsymbol{X}}^t(a_l) &= \tilde{\boldsymbol{X}}^t(a_1)\tilde{\boldsymbol{X}}^t(a_2)\cdots\tilde{\boldsymbol{X}}^t(a_q)\tilde{\boldsymbol{X}}^t(a_l) \\
&= \alpha'\boldsymbol{v}(a_1)\boldsymbol{v}(a_q)^\top\tilde{\boldsymbol{X}}^t(a_l) \\
&= \alpha'\boldsymbol{v}(a_1)\boldsymbol{v}(a_q)^\top\boldsymbol{v}(a_l)\boldsymbol{v}(a_l)^\top \\
&= (\boldsymbol{v}(a_q)^\top\boldsymbol{v}(a_l))\,\alpha'\boldsymbol{v}(a_1)\boldsymbol{v}(a_l)^\top \\
&= \left(\frac{\mathcal{M}^+(i_{a_q}, i_{a_l})}{2\mathcal{R}_M} + \frac{\mathcal{N}^+(j_{a_q}, j_{a_l})}{2\mathcal{R}_N}\right)\alpha'\boldsymbol{v}(a_1)\boldsymbol{v}(a_l)^\top
\end{aligned}
$$

Similarly we have:

$$
\tilde{\boldsymbol{X}}^{T+1}(a_1)\tilde{\boldsymbol{X}}^{T+1}(a_2)\cdots\tilde{\boldsymbol{X}}^{T+1}(a_l) = \left(\frac{\mathcal{M}^+(i_{a_q}, i_{a_l})}{2\mathcal{R}_M} + \frac{\mathcal{N}^+(j_{a_q}, j_{a_l})}{2\mathcal{R}_N}\right)\alpha'\bar{\boldsymbol{v}}(a_1)\bar{\boldsymbol{v}}(a_l)^\top,
$$

from which the result follows. $\qquad\square$

**Lemma 31.** *For all $l \in \mathbb{N}$ and for all $a_1, a_2, ..., a_l \in \mathbb{U} \cap [t-1]$ we have:*

$$
\mathrm{tr}\left(\tilde{\boldsymbol{X}}^t(t)\tilde{\boldsymbol{X}}^t(a_1)\tilde{\boldsymbol{X}}^t(a_2)\cdots\tilde{\boldsymbol{X}}^t(a_l)\right) = \mathrm{tr}\left(\tilde{\boldsymbol{X}}^{T+1}(t)\tilde{\boldsymbol{X}}^{T+1}(a_1)\tilde{\boldsymbol{X}}^{T+1}(a_2)\cdots\tilde{\boldsymbol{X}}^{T+1}(a_l)\right)
$$

*Proof.* By Lemma 30, let $\alpha$ be such that

$$
\tilde{\boldsymbol{X}}^t(a_1)\tilde{\boldsymbol{X}}^t(a_2)\cdots\tilde{\boldsymbol{X}}^t(a_l) = \alpha\boldsymbol{v}(a_1)\boldsymbol{v}(a_l)^\top
$$

and

$$
\tilde{\boldsymbol{X}}^{T+1}(a_1)\tilde{\boldsymbol{X}}^{T+1}(a_2)\cdots\tilde{\boldsymbol{X}}^{T+1}(a_l) = \alpha\bar{\boldsymbol{v}}(a_1)\bar{\boldsymbol{v}}(a_l)^\top.
$$

Note that:

$$
\begin{aligned}
&\mathrm{tr}\left(\tilde{\boldsymbol{X}}^t(t)\tilde{\boldsymbol{X}}^t(a_1)\tilde{\boldsymbol{X}}^t(a_2)\cdots\tilde{\boldsymbol{X}}^t(a_l)\right) \\
&= \alpha\,\mathrm{tr}\left(\tilde{\boldsymbol{X}}^t(t)\boldsymbol{v}(a_1)\boldsymbol{v}(a_l)^\top\right) \\
&= \alpha\,\mathrm{tr}\left(\boldsymbol{v}(t)\boldsymbol{v}(t)^\top\boldsymbol{v}(a_1)\boldsymbol{v}(a_l)^\top\right) \\
&= \alpha\,\mathrm{tr}\left(\boldsymbol{v}(a_l)^\top\boldsymbol{v}(t)\boldsymbol{v}(t)^\top\boldsymbol{v}(a_1)\right) \\
&= \alpha\left(\boldsymbol{v}(a_l)^\top\boldsymbol{v}(t)\right)\left(\boldsymbol{v}(t)^\top\boldsymbol{v}(a_1)\right) \\
&= \alpha\left(\frac{\mathcal{M}^+(i_{a_l}, i_t)}{2\mathcal{R}_M} + \frac{\mathcal{N}^+(j_{a_l}, j_t)}{2\mathcal{R}_N}\right)\left(\frac{\mathcal{M}^+(i_t, i_{a_1})}{2\mathcal{R}_M} + \frac{\mathcal{N}^+(j_t, j_{a_1})}{2\mathcal{R}_N}\right)
\end{aligned}
$$

Similarly we have:

$$
\mathrm{tr}\left(\tilde{\boldsymbol{X}}^{T+1}(t)\tilde{\boldsymbol{X}}^{T+1}(a_1)\tilde{\boldsymbol{X}}^{T+1}(a_2)\cdots\tilde{\boldsymbol{X}}^{T+1}(a_l)\right) \tag{49}
$$

$$
= \alpha\left(\frac{\mathcal{M}^+(i_{a_l}, i_t)}{2\mathcal{R}_M} + \frac{\mathcal{N}^+(j_{a_l}, j_t)}{2\mathcal{R}_N}\right)\left(\frac{\mathcal{M}^+(i_t, i_{a_1})}{2\mathcal{R}_M} + \frac{\mathcal{N}^+(j_t, j_{a_1})}{2\mathcal{R}_N}\right) \tag{50}
$$

The result follows. $\qquad\square$

**Lemma 32.** *For any $q \in \mathbb{N}$, any $\kappa \in \Re^+$ and any $b_1, b_2, \cdots b_{t-1} \in \Re$ we have:*

$$
\mathrm{tr}\left(\tilde{\boldsymbol{X}}^t(t)\left(\sum_{s \in \mathbb{U} \cap [t-1]} b_s\tilde{\boldsymbol{X}}^t(s)\right)^q\right) = \mathrm{tr}\left(\tilde{\boldsymbol{X}}^{T+1}(t)\left(\sum_{s \in \mathbb{U} \cap [t-1]} b_s\tilde{\boldsymbol{X}}^{T+1}(s)\right)^q\right)
$$

*Proof.* We have:

$$\text{tr}\left(\tilde{\boldsymbol{X}}^t(t)\left(\sum_{s\in\mathbb{U}\cap[t-1]}b_s\tilde{\boldsymbol{X}}^t(s)\right)^q\right)$$

$$= \text{tr}\left(\tilde{\boldsymbol{X}}^t(t)\sum_{a_1\in\mathbb{U}\cap[t-1]}\sum_{a_2\in\mathbb{U}\cap[t-1]}\cdots\sum_{a_q\in\mathbb{U}\cap[t-1]}\left(\prod_{i=1}^q b_{a_i}\right)\tilde{\boldsymbol{X}}^t(a_1)\tilde{\boldsymbol{X}}^t(a_2)\cdots\tilde{\boldsymbol{X}}^t(a_q)\right)$$

$$= \sum_{a_1\in\mathbb{U}\cap[t-1]}\sum_{a_2\in\mathbb{U}\cap[t-1]}\cdots\sum_{a_q\in\mathbb{U}\cap[t-1]}\left(\prod_{i=1}^q b_{a_i}\right)\text{tr}\left(\tilde{\boldsymbol{X}}^t(t)\tilde{\boldsymbol{X}}^t(a_1)\tilde{\boldsymbol{X}}^t(a_2)\cdots\tilde{\boldsymbol{X}}^t(a_q)\right)$$

and similarly,

$$\text{tr}\left(\tilde{\boldsymbol{X}}^{T+1}(t)\left(\sum_{s=1}^{t-1}b_s\tilde{\boldsymbol{X}}^{T+1}(s)\right)^q\right)$$

$$= \text{tr}\left(\tilde{\boldsymbol{X}}^{T+1}(t)\sum_{a_1\in\mathbb{U}\cap[t-1]}\sum_{a_2\in\mathbb{U}\cap[t-1]}\cdots\sum_{a_q\in\mathbb{U}\cap[t-1]}\left(\prod_{i=1}^q b_{a_i}\right)\tilde{\boldsymbol{X}}^{T+1}(a_1)\tilde{\boldsymbol{X}}^{T+1}(a_2)\cdots\tilde{\boldsymbol{X}}^{T+1}(a_q)\right)$$

$$= \sum_{a_1\in\mathbb{U}\cap[t-1]}\sum_{a_2\in\mathbb{U}\cap[t-1]}\cdots\sum_{a_q\in\mathbb{U}\cap[t-1]}\left(\prod_{i=1}^q b_{a_i}\right)\text{tr}\left(\tilde{\boldsymbol{X}}^{T+1}(t)\tilde{\boldsymbol{X}}^{T+1}(a_1)\tilde{\boldsymbol{X}}^{T+1}(a_2)\cdots\tilde{\boldsymbol{X}}^{T+1}(a_q)\right).$$

The result follows by Lemma 31. $\qquad\square$

**Lemma 33.** *For any $\kappa\in\Re^+$ and any $b_1,b_2,\cdots b_{t-1}\in\Re$ we have:*

$$\text{tr}\left(\tilde{\boldsymbol{X}}^t(t)\exp\left(\kappa\boldsymbol{I}+\sum_{s\in\mathbb{U}\cap[t-1]}b_s\tilde{\boldsymbol{X}}^t(s)\right)\right) = \text{tr}\left(\tilde{\boldsymbol{X}}^{T+1}(t)\exp\left(\kappa\boldsymbol{I}+\sum_{s\in\mathbb{U}\cap[t-1]}b_s\tilde{\boldsymbol{X}}^{T+1}(s)\right)\right)$$

*Proof.* Using the fact that $\exp(\boldsymbol{A}+\boldsymbol{B})=\exp(\boldsymbol{A})\exp(\boldsymbol{B})$ for commuting matrices $\boldsymbol{A}$ and $\boldsymbol{B}$, and noting that the multiple of the identity matrix commutes with any matrix, we have that

$$\text{tr}\left(\tilde{\boldsymbol{X}}^t(t)\exp\left(\kappa\boldsymbol{I}+\sum_{s\in\mathbb{U}\cap[t-1]}b_s\tilde{\boldsymbol{X}}^t(s)\right)\right) = \text{tr}\left(\tilde{\boldsymbol{X}}^t(t)\exp(\kappa\boldsymbol{I})\exp\left(\sum_{s\in\mathbb{U}\cap[t-1]}b_s\tilde{\boldsymbol{X}}^t(s)\right)\right).$$

By the Taylors series expansion we have:

$$\text{tr}\left(\tilde{\boldsymbol{X}}^t(t)\exp(\kappa\boldsymbol{I})\exp\left(\sum_{s\in\mathbb{U}\cap[t-1]}b_s\tilde{\boldsymbol{X}}^t(s)\right)\right) = e^\kappa\,\text{tr}\left(\tilde{\boldsymbol{X}}^t(t)\sum_{q=0}^\infty\frac{1}{q!}\left(\sum_{s\in\mathbb{U}\cap[t-1]}b_s\tilde{\boldsymbol{X}}^t(s)\right)^q\right)$$

$$= e^\kappa\sum_{q=0}^\infty\frac{1}{q!}\text{tr}\left(\tilde{\boldsymbol{X}}^t(t)\left(\sum_{s\in\mathbb{U}\cap[t-1]}b_s\tilde{\boldsymbol{X}}^t(s)\right)^q\right)$$

Similarly, we have

$$\text{tr}\left(\tilde{\boldsymbol{X}}^{T+1}(t)\exp\left(\kappa\boldsymbol{I}+\sum_{s\in\mathbb{U}\cap[t-1]}b_s\tilde{\boldsymbol{X}}^{T+1}(s)\right)\right) = e^\kappa\sum_{q=0}^\infty\frac{1}{q!}\text{tr}\left(\tilde{\boldsymbol{X}}^{T+1}(t)\left(\sum_{s\in\mathbb{U}\cap[t-1]}b_s\tilde{\boldsymbol{X}}^{T+1}(s)\right)^q\right).$$

The result then follows from Lemma 32. $\qquad\square$

### D.1.2 Equivalence of Algorithms

On a trial $t$ let $\bar{z}_t$ be the prediction $(\bar{y}_t)$ of the inductive algorithm and let $\bar{y}_t$ remain the prediction of the transductive algorithm. We fix $\kappa := \log\left(\widehat{\mathcal{D}}/(m+n)\right)$.

**Lemma 34.** *On a trial $t$ the prediction, $\bar{y}_t$, of the transductive algorithm is given by:*

$$\bar{y}_t = \operatorname{tr}\left(\tilde{\boldsymbol{X}}^{T+1}(t)\exp\left(\kappa\boldsymbol{I} + \sum_{s=1}^{t-1}f_s(\bar{y}_s)\tilde{\boldsymbol{X}}^{T+1}(s)\right)\right)$$

*and the prediction, $\bar{z}_t$, of the inductive algorithm is given by:*

$$\bar{z}_t = \operatorname{tr}\left(\tilde{\boldsymbol{X}}^t(t)\exp\left(\kappa\boldsymbol{I} + \sum_{s=1}^{t-1}f_s(\bar{z}_s)\tilde{\boldsymbol{X}}^t(s)\right)\right)$$

*where $f_s(x) := \eta y_s$ if $y_s x \leq [\text{NON-CONSERVATIVE}]\times\gamma$ and $f_s(x) := 0$ otherwise.*

*Proof.* Direct from algorithms, noting that if $s \notin \mathbb{U}\cap[t-1]$ then $f_s(\bar{z}_s) = 0$. □

**Lemma 35.** *Given a trial $t$, if $\bar{y}_s = \bar{z}_s$ for all $s < t$, then $\bar{y}_t = \bar{z}_t$.*

*Proof.* Direct from Lemmas 34 and 33 (with $b_s := f_s(\bar{y}_t) = f_s(\bar{z}_t)$), noting that if $s \notin \mathbb{U}\cap[t-1]$ then $f_s(\bar{z}_s) = 0$. □

Proposition 4 follows by induction over Lemma 35. ∎

### D.2 Proof of Proposition 5

In the following, we define $\mathcal{K}_{\boldsymbol{x}}(\cdot) := \mathcal{K}(\boldsymbol{x}, \cdot)$. If $r \geq 2$, $\delta^* := \min\left(2, \frac{1}{4}\delta(S_1, \ldots, S_k)\right)$. This implies that $\delta^* \leq \min\left(2, \frac{r-1}{2r}\delta(S_1, \ldots, S_k)\right)$. Recall that $s(\boldsymbol{x}) := \frac{r-1}{2r}\boldsymbol{x} + \frac{r+1}{2}\boldsymbol{1}$. Then observe that, given that the transformation $\tilde{\boldsymbol{x}}_i = s(\boldsymbol{x}_i)$ holds true for all $i \in [m]$, requiring $S_1, \ldots, S_k \subset [-r, r]^d$ with $\boldsymbol{x}_1, \ldots, \boldsymbol{x}_m \in \cup_{i=1}^k S_i$ and $\delta^* \leq \min\left(2, \frac{r-1}{2r}\delta(S_1, \ldots, S_k)\right)$ is equivalent to the requirement that $\tilde{S}_1, \ldots, \tilde{S}_k \subset [1, r]^d$ with $\tilde{\boldsymbol{x}}_1, \ldots, \tilde{\boldsymbol{x}}_m \in \cup_{i=1}^k \tilde{S}_i$ and $\delta^* \leq \min\left(2, \delta(\tilde{S}_1, \ldots, \tilde{S}_k)\right)$. Furthermore, for all $i \in [m]$ and $j \in [k]$, we have that $\boldsymbol{x}_i \in S_j$ if and only if $\tilde{\boldsymbol{x}}_i \in \tilde{S}_j$. We shall proceed with the latter set of requirements for simplicity. Recall that the RKHS for the $d = 1$ min kernel $\mathcal{H}_{\mathcal{K}}^1$ is the set of all absolutely continuous functions from $[0, \infty)^d \to \Re$ that satisfy $f(0) = 0$ and $\int_0^\infty [f'(x)]^2 \mathrm{d}x < \infty$.

**Lemma 36.** *The inner product for $f \in \mathcal{H}_{\mathcal{K}}^1$ may be computed by,*

$$\langle f, g\rangle = \int_0^\infty f'(x)g'(x)\mathrm{d}x\,.$$

*Proof.* We show this by the reproducing property:

$$\langle f, \mathcal{K}_x\rangle = f(x).$$

Defining $\boldsymbol{1}_x(t)$ as the step function that evaluates to 1 for $t \leq x$ and 0 otherwise, we note that the derivative of $\min(x, t)$ with respect to $t$ is equal to $\boldsymbol{1}_x(t)$. This gives rise to

$$\int_0^\infty f'(t)\mathcal{K}'(x, t)\mathrm{d}t = \int_0^\infty f'(t)\boldsymbol{1}_x(t)\mathrm{d}t = \int_0^x f'(t)\mathrm{d}t = f(x)\,.$$

Using the condition of $f(0) = 0$, we then obtain the reproducing property. □

**Lemma 37.** *Given $k$ boxes $\tilde{S}_1, \ldots, \tilde{S}_k \subset [1, r]^d$, $\delta^* \leq \min\left(2, \delta(\tilde{S}_1, \ldots, \tilde{S}_k)\right)$ and $\tilde{\boldsymbol{x}}_1, \ldots, \tilde{\boldsymbol{x}}_m \in \cup_{i=1}^k \tilde{S}_i$, there exists a function $f \in H_{\mathcal{K}}$ for which $f(\tilde{\boldsymbol{x}}_j) = [\tilde{\boldsymbol{x}}_j \in \tilde{S}_1]$ for $j \in [m]$ and this function has norm*

$$||f||^2 = \left(\frac{4}{\delta^*}\right)^d\,.$$

Figure 3: Visualization of the function $f(x_1, x_2)$ with $S_1$, $S_2$ and $S_3$ represented as red rectangles in the $x_1 - x_2$ plane.

*Proof.* Recall that a *box* in $\Re^d$ is a set $\{\boldsymbol{x} : a_i \leq x_i \leq b_i, i \in [d]\}$ defined by a pair of vectors $\boldsymbol{a}, \boldsymbol{b} \in \Re^d$. First, we consider the case of $d = 1$, with the coordinates of $\tilde{S}_1$ defined by $a$ and $b$. Defining the function that interpolates the points $\tilde{\boldsymbol{x}}_1, \dots \tilde{\boldsymbol{x}}_m$ in one dimension as $f^1 \in \mathcal{H}_{\mathcal{K}}^1$, we chose $f^1$ to be the following:

$$
f^1(x) = \begin{cases}
0 & \text{for} \quad x \leq a - \frac{\delta^*}{2} \\
\frac{2}{\delta^*} x + 1 - \frac{2}{\delta^*} a & \text{for} \quad a - \frac{\delta^*}{2} < x \leq a \\
1 & \text{for} \quad a < x \leq b \\
-\frac{2}{\delta^*} x + 1 + \frac{2}{\delta^*} b & \text{for} \quad b < x \leq b + \frac{\delta^*}{2} \\
0 & \text{for} \quad x > b + \frac{\delta^*}{2}.
\end{cases}
$$

This function is picked from the space $\mathcal{H}_{\mathcal{K}}^1$ so that $\int_0^\infty [(f^1)'(x)]^2 \mathrm{d}x$ is minimized with respect to "worst-case" constraints. The condition on $\delta^*$ implies that $\delta^* \leq 2$, so that $f^1(0) = 0$. It also implies that $\delta^* \leq \delta(S_1, \dots S_k)$ so that for all $i \in [m]$, $f^1(\tilde{x}_i) = 0$ if $\tilde{x}_i \notin S_1$. The norm $||f^1||^2$, then becomes

$$
||f^1||^2 = \int_0^\infty |(f^1)'(x)|^2 \mathrm{d}x
$$

$$
= \int_{a-\frac{\delta^*}{2}}^a \left(\frac{2}{\delta^*}\right)^2 \mathrm{d}x + \int_b^{b+\frac{\delta^*}{2}} \left(\frac{2}{\delta^*}\right)^2 \mathrm{d}x
$$

$$
= 2\left(\frac{2}{\delta^*}\right)^2 \left(\frac{\delta^*}{2}\right) = \frac{4}{\delta^*}.
$$

This can be extended to multiple dimensions by observing that the induced product norm of $f$ is the product of the norms of $f^1$ in each dimension, thus giving the required bound. In this case also, the condition on $\delta^*$ ensures both $f(\mathbf{0}) = 0$ and $f(\tilde{\boldsymbol{x}}_i) = 0$ for $\tilde{\boldsymbol{x}}_i \notin \tilde{S}_1$, where $i \in [m]$. For an illustration of this function in two dimensions, see Figure 3. □

**Lemma 38.** *Given $k$ boxes $\tilde{S}_1, \dots, \tilde{S}_k \subset [1, r]^d$, $\delta^* \leq \min\left(2, \delta(\tilde{S}_1, \dots, \tilde{S}_k)\right)$ and $\tilde{\boldsymbol{x}}_1, \dots, \tilde{\boldsymbol{x}}_m \in \cup_{i=1}^k \tilde{S}_i$, if $\boldsymbol{u} = (u_i = [\tilde{\boldsymbol{x}}_i \in \tilde{S}_1])_{i \in [m]}$ and $\boldsymbol{K} = (\mathcal{K}(\tilde{\boldsymbol{x}}_i, \tilde{\boldsymbol{x}}_j))_{i,j \in [m]}$ then $\boldsymbol{u}^\top \boldsymbol{K}^{-1} \boldsymbol{u} \leq \left(\frac{4}{\delta^*}\right)^d$.*

*Proof.* Using Lemma 37 we observe that,

$$
\boldsymbol{u}^\top \boldsymbol{K}^{-1} \boldsymbol{u} = \underset{f \in H_{\mathcal{K}} : f(\tilde{\boldsymbol{x}}_i) = [\tilde{\boldsymbol{x}}_i \in \tilde{S}_1], i \in [m]}{\operatorname{argmin}} ||f||_{\mathcal{K}}^2 \leq \underset{f \in H_{\mathcal{K}} : f(\tilde{\boldsymbol{x}}) = [\tilde{\boldsymbol{x}} \in \tilde{S}_1], \tilde{\boldsymbol{x}} \in \cup_{i \in [k]} \tilde{S}_i}{\operatorname{argmin}} ||f||_{\mathcal{K}}^2 \leq \left(\frac{4}{\delta^*}\right)^d,
$$

for $\boldsymbol{u} := (u_i = [\tilde{\boldsymbol{x}}_i \in \tilde{S}_1])_{i \in [m]}$, $\boldsymbol{K} := (\mathcal{K}(\tilde{\boldsymbol{x}}_i, \tilde{\boldsymbol{x}}_j))_{i,j \in [m]}$ and $\tilde{\boldsymbol{x}}_1, \ldots, \tilde{\boldsymbol{x}}_m \in \cup_{i \in [k]} \tilde{S}_i$. Note that the second term in the equation above has a constraint in terms of the given $m$ points $\tilde{\boldsymbol{x}}_1, \ldots, \tilde{\boldsymbol{x}}_m$, whereas the optimization in the third term has a similar constraint, but in terms of any $\tilde{\boldsymbol{x}}$ that satisfies $\tilde{\boldsymbol{x}} \in \cup_{i \in [k]} \tilde{S}_i$. $\qquad\square$

Defining $\boldsymbol{u}_i = (\boldsymbol{R}_i^\top)^\top$, then observe that the term $\mathrm{tr}(\boldsymbol{R}^\top \boldsymbol{K}^{-1} \boldsymbol{R}) = \sum_{i \in [k]} \boldsymbol{u}_i^\top \boldsymbol{K}^{-1} \boldsymbol{u}_i$. Thus by applying Lemma 38 to each $\boldsymbol{u}_i$, we have that $\mathrm{tr}(\boldsymbol{R}^\top \boldsymbol{K}^{-1} \boldsymbol{R}) \le k(\frac{4}{\delta^*})^d$. $\qquad\blacksquare$