[Reviews · NeurIPS 2020]

Review 1

Summary and Contributions: This paper incorporates side information in the form of row and column similarity matrices for online matrix completion. They propose an algorithm which builds upon matrix exponentiated gradient, and provide two variants for transductive models and inductive models. Transductive assumes the full side information matrices are known in advance, and inductive assumes the kernel function is known, but the realization of the latent variables associated to the row and column pairs is only revealed online. Authors provide regret bounds in the adversarial worst case setting for both transductive and inductive settings, and show how the bounds adapt to specialized structures such as latent block structures in the form of a bi-clustered matrix, graph-based information, and community structure.

Strengths: Incorporating side information is a practical and valuable extension to online matrix completion. The results specializing to specific structures are helpful, primarily coming into play by bounding the quasi-dimension of the matrix relative to the side information. Results seem correct as far as I can tell.

Weaknesses: The algorithm is a modification of Matrix Exponentiated Gradient, but there is some missing discussion to provide intuition for the form of how side information is taken into consideration and how that changes the analysis/proof. There is no explanation of what the "non-conservative" flag is used for and why one might choose one or the other value depending on the application.

Correctness: For Theorem 1, I am confused why regret depends on gamma when non-conservative = 0? In the pseudo-code of the algorithm, when non-conservative is set to 0, then there is no noise added, so the algorithm results don't depend on the value of gamma at all, so why does it show up in the bounds?

Clarity: The paper could have more background about matrix exponentiated gradient as it forms the basis of the algorithm and results they present. It would be helpful to explain how the new modifications affect the analysis (some abbreviated proof sketch would be appreciated), as the results are only stated with no discussion about what it takes to extend the previous analysis for this setting.

Relation to Prior Work: It is clear that the novelty is incorporating similarity matrix side information, but the algorithm is a modification upon MEG with no explanation of how the modification changes the analysis (e.g. were any new proof techniques required).

Reproducibility: Yes

Additional Feedback: Font size for algorithm statement is waaayyy too tiny.


Review 2

Summary and Contributions: This paper gives theoretical guarantees on online binary matrix completion with side information. Compared to the normally low-rank assumption, the paper makes assumption of “latent block structure”, where each row/column are associated with a label/cluster, and the entries are totally determined by the labels/clusters of both row and column. The paper proposes two algorithms for the problem: both transductive and inductive. Then the (expected) number of mistakes made compared to perfect one are given. The bound depends on two properties: how the side information help, and the latent structure of the matrix (similar to the underlying rankness).

Strengths: The paper is the first paper to study “online binary matrix completion with side information” problem from a theoretical point of view. The paper provides solid bounds for the online binary matrix completion with side information problem. The paper discusses the two key factors in the bounds, and how to bound them.

Weaknesses: The title is not appropriate. It actually talks about “binary matrix completion” instead of classical matrix completion. (this problem is easy to be revised.) The paper is not reader-friendly. There is a lot of thing unexplained in the paper. For example, what is the motivation for the two proposed algorithms, and why they are reasonable to solve the current problem is not discussed. The paper directly rushes to the theoretical results to bound the mistakes. What does “non-conservative” and “conservative” mean in the paper is never explained. Why does mc(U) called the margin complexity, and how it resembles the margin in SVM or other classical machine learning algorithms? The paper uses nearly one-whole page to define the notations used in the paper. I am afraid it may not be an appropriate way to assign so many space instead of giving a clear picture of the intuition meaning of the notation/algorithm. It is better (and highly suggested) to add explanations why some of the notations are proposed. One example is SP(U). Is it proposed because of we are doing binary classification so only the sign of the entries are important but not the exact value? Given the complex notation system, I think it is better to 1) group the notations, and use one common symbol for the same group in the text, and put detailed explanation in the supplementary; 2) add intuitive explanations why we need such notations. Given a large audience of machine learning conferences these years, we may need to write a paper to draw people’s attention, instead of scare them. The “latent block structure” seems not well-motivated from a practical part. In Netflix, it should be a five rated problem, instead of a binary rated problem. From this point of view, the latent block structure may be a strong assumption and not easy to be tested in reality. --------------------------------------------- Thanks for the rebuttal. However, not all of my concerns have been clarified. I will keep my score.

Correctness: The claims are correct. The method is correct from theoretical aspects. But why the procedure is correct is never explained in the paper.

Clarity: No. Please refer to weaknesses.

Relation to Prior Work: The paper is the first paper to study such a problem. It is hard to compare its results directly with previous paper. The current paper summarized previous work, and slightly compared their results.

Reproducibility: Yes

Additional Feedback:


Review 3

Summary and Contributions: This work deals with the problem of Online Matrix Prediction with side information, both in the transductive and inductive (online) models. The bounds are given in terms of two quantities: (1) the margin complexity of the comparator matrix U, a notion introduced by Ben David et al. in “Learning Sparse Low-Threshold Linear Classifiers”; (2) the quasi-dimension, a concept introduced in this work which quantifies how much the side information reveals about the comparator matrix. The main algorithmic ingredient is an adaptation of the algorithm introduced in “Learning Sparse Low-Threshold Linear Classifiers”, which is, in turn, an instance of the well-known EG algorithm. The challenge is to translate the algorithm to the matrix setting and provide a regret bound, along the lines of what was done before in similar settings. The main novelty relies in relating the bounds to the quasi-dimension. The key assumptions are known upper bounds on the quasi-dimension and on the margin complexity for tuning the parameters of the algorithm and derive a mistake bound. The authors also provide some examples when these parameters might be known, for instance when the competitor matrix U is a binary-clustered matrix. In the graph-based side information case of the transductive setting, and under the additional assumption that the partition which maps rows to factors is known, the authors show improvements when using side-information. Also, they are able to extend and recover previous results on "similarity" prediction. On the other hand, in the more difficult inductive setting (i.e., when the side information is revealed over time), the authors derive a kernelized variant of the algorithm.

Strengths: This work introduces several notions which might be interesting for future works in related settings, such as the formulation of side information for matrix completion, the "quasi-dimension" and the online inductive setting. The proposed algorithms are carefully analysed in different applications and upper bounds on their regret are provided. The proof is based on the Follow-the-Regularized-Leader framework (with entropic regularisation) but its extension to the setting considered is not straightforward.

Weaknesses: Lower bounds are somehow discussed only informally and it is not totally clear what is possible to achieve in the setting introduced by the authors. Finally, the running time of the proposed algorithms is prohibitive for real-world application, as also highlighted on the experiments in the appendix (even though I suppose this was not the main point of the paper). **After rebuttal**: I share other reviewers' concern about the clarity of this paper. The technical contribution is relevant, however the results are not well presented and easy to understand.

Correctness: All the proofs are sound and well-written.

Clarity: The paper is notationally heavy and sometimes hard to follow. I also think a discussion in the end summarising the main results could be beneficial.

Relation to Prior Work: Yes, it is.

Reproducibility: Yes

Additional Feedback:


Review 4

Summary and Contributions: The paper proposes two algorithms for online matrix completion in the presence of side information. This side information is either available as two matrices that specify similarity among rows and columns (inductive setting) or two kernels that can construct these similarities on the go (transductive setting). The paper proves regret and mistake bounds for the proposed algorithms. These bounds depend on what the authors call a quasi-dimension. The paper further proves bounds on quasi-dimension in specialized settings where the underlying true label matrix has a latent block structure and where the side information is continuous but contained in k disjoint boxes in R^d. Online community membership prediction is also included as an application. The main contributions are: Proposed and analyzed online matrix completion algorithms in the inductive and transductive settings to prove bounds on the number of mistakes and regret. Showed example settings where the quasi-dimension used in the bounds above can be easily bounded.

Strengths: 1. The paper studies an interesting problem in the online setting, especially in the inductive case. 2. Useful bounds have been shown for important cases like the matrix with latent block structure.

Weaknesses: 1. The biggest issue I have with the paper is its clarity: (a) The presentation can be improved. (b) More intuition about the algorithms can be provided. (c) The exposition in certain places, especially in Section 4.1, can be simplified (it would be easier to understand the structure of matrices M and N directly in Section 4.1 rather than arriving at them from a graph perspective). (d) Terms like margin complexity, comparator matrix, and so on appear in the introduction without being defined first. (e) The related works section contains technical details that make it hard to understand unless one has read the entire paper. However, this section is placed before any such details have been discussed. (f) What are R_M and R_N in (3)? R_L in the paragraph before algortihms? The iff statement after (3) needs more justification. (g) Concluding remarks are not given. (2) The transductive setting seems unnatural. Why is it reasonable to assume that M and N will be available beforehand? (3) Experiments on some real data would have been useful. (4) The time complexity of the algorithm seems to be prohibitive. In the absence of experiments with real data, it is not easy to understand how many steps will be typically required. It is important to understand this as each step is very expensive.

Correctness: More clarity on the following points would be welcome: I find it counterintuitive that mistakes and regret are inversely proportional to gamma. Shouldn't more mistakes be made when the margin requirements are higher? How is the max norm block invariant? I believe that as the sizes of the matrices are different, they will be on different scales. Where is the minimum in (7)?

Clarity: I have several issues with the clarity of this paper. More details are given above.

Relation to Prior Work: I feel that the authors have done a good job of putting their work in an appropriate context. However, I am unsure about the novelty of contributions.

Reproducibility: Yes

Additional Feedback: See above. ***Thanks for the rebuttal.

[Author Response · NeurIPS 2020]

**R1:** *"non-conservative flag"* : In **practice** the algorithm should **always** be run non-conservatively to obtain a regret bound as that is the more flexible/noise-tolerant bound. We **only** include the conservative case for presentation issues as the bound is *simpler*, analogous to Novikoff's (perceptron) bound, and in order to give "easy" upper and lower bound insights in that case (e.g., lines 244-249, 350-353). We will update the manuscript to reflect this advice.

*" For Theorem 1, ... non-conservative = 0? ... don't depend on the value of $\gamma$ at all, so why does it show up in the bounds?"* : See line 172 - 173, 1) the algorithm's predictions depends on $\gamma$ through $\eta$ ($\eta = \gamma$). 2) the bound fails hold if for the "true" comparator matrix $U$ the margin complexity is large i.e., mc$(U) > 1/\gamma$.

*"modification upon MEG ... (e.g. were any new proof techniques required)."* : See 109-112 which then refers to Appendix B.2 (esp. lines 520-536) which discusses our new proof techniques wrt MEG.

**R2:** *"Why does mc$(U)$ ... resembles the margin in SVM"*: In lines 2-8 of the abstract we discuss the term $1/\gamma^2$ which serves as the algorithmic proxy for mc$^2(U)$ i.e., the tightest bound is obtained if $1/\gamma^2 = $ mc$^2(U)$ and it is in those lines that we make the explicit connection to the "SVM margin." This initial discussion is then built on in lines 54-61 where we further refer to the three references [5,6,7].

*"The latent block structure seems not well-motivated from a practical part. In Netflix, it should be a five rated problem ... , the latent block structure may be a strong assumption and not easy to be tested in reality."* : See lines 40-53 (esp. 51-53). Observe that the latent block structure (LBS) assumption is actually a *weaker* assumption than the common low-rank assumption. Note that Netflix does not meet the low-rank assumption as it is experimentally known that human ratings are only *ordinally* qualitative. Finally the definition of LBS trivially generalizes to categorical data as do our algorithms.

**R3:** *"Lower bounds"* : We agree that there is value in formalizing the setting. Our casual mistake lower bounds generalize to lower bounds for regret using [*Agnostic Online Learning*, 2009, Ben-David et al; Lem. 14].

**R4:** *"(c) The exposition ... Section 4.1, can be simplified ... matrices M and N directly in Section 4.1 rather than arriving at them from a graph perspective).* Yes, an alternate presentation is to assume feature vectors associated with each row and column and let $M$ and $N$ be the inverse of the gram (kernel) matrices (also see 251-256). However we focus on the graph perspective because of the smaller bound on $\mathcal{D}$ wrt graphs given in Thm 3. Eq (7). We note in the batch setting different methods of using graph side information were considered in [15,16] (lines 87-94). In the inductive setting (Sec. 5.1) we give an example of using non-graph side-information. Finally we note that 4.1 serves as the necessary background for 4.2 which has two important observations: 1) a bound from [22] can be recovered and extended, 2) An example of a class of $k \times k$ biclustered matrices (whose rank are $k$) for which max norm is $O(1)$.

*"(d), (e)"* : We will move related work and use forward references for undefined notation.
*" (f) What are $\mathcal{R}_M, \mathcal{R}_N, \mathcal{R}_L$ ..."* : See line 129. The *"Iff"* (147-148) .. follows from definition of $\|U\|_{\max}$.

*"(2) The transductive setting seems unnatural. Why is it reasonable to assume that $M$ and $N$ will be available beforehand?"* : We argue side-information is the norm rather than the exception. I.e., in the Netflix example we may have demographic info on the *users* as well as categorization, actor lists, etc on the *movies*. $M$ and $N$ may then be constructed from feature vectors by selecting kernels and inverting the kernel matrices. Or as in graph-based semi-supervised learning, a graph may be constructed using the feature vectors and using the corresponding Laplacian.

*"(3) ... experiments (4) ...time complexity ..."* : We agree that experiments would be useful and that the time complexity is large. However, we note that natural heuristics include maintaining only a low-rank approximation to $\tilde{W}^t$ and maintaining a fixed number of indices in $\mathbb{U}$ (e.g., decaying old indices) for Algs. 1 and 2, respectively.

*"... counterintuitive ... inversely proportional to gamma. ... Shouldn't more mistakes be made when the margin requirements are higher?"* : See lines 3-4 of the Abstract. This is the usual intuition behind perceptron, SVM, and other "largin margin" classifiers. I.e., the further that data in classes are apart the easier it is separate and thus (online) fewer mistakes (batch) better generalization as opposed to the case where the classes are arbitrarily near one another.

*"How is the max norm block invariant? I believe that as the sizes of the matrices are different, they will be on different scales."* : **Theorem: max-norm is block invariant.** *Pf. Sketch.* We first show $\|X\|_{\max} \geq \|RXC^\top\|_{\max}$ for all $m, k, n, \ell \in \mathbb{N}^+$ with $m \geq k$, $n \geq \ell$, $R \in \mathcal{B}^{m,k}$ and $C \in \mathcal{B}^{n,\ell}$ (where $\mathcal{B}^{m,k}$ is the set of all $m \times k$ block expansion matrices (cf lines 140-143)). WLOG. assume $X$ is $k \times \ell$. If $PQ^\top = X$ then $(RP)(CQ)^\top = RXC^\top$. Observe that $\max_{i \in [k]} \|P_i\| = \max_{s \in [m]} \|(RP)_s\|$ since every row in $P$ is duplicated by (1+) rows in $(RP)$ and there are no distinct rows in $(RP)$ that are not in $P$. Recall $\|X\|_{\max} := \min_{PQ^\top = X}\{\max_{1 \leq i \leq m} \|P_i\| \times \max_{1 \leq j \leq n} \|Q_j\|\}$ then since for every decomposition $PQ^\top = X$ there exists a decomposition $(RP)(CQ)^\top = RXC^\top$ thus $\|X\|_{\max} \geq \|RXC^\top\|_{\max}$. We have $\|X\|_{\max} \leq \|RXC^\top\|_{\max}$ since trivially the max-norm of a sub-matrix cannot be larger than that of the matrix ∎.

"Where is the minimum in (7)?" : See line 205. The minimum is before the large '{' in (7) and is over the matrices $R, C, U^*$ s.t. $U = RU^*C^\top$.

[Meta-Review · NeurIPS 2020]

Contributions: As nicely summarized by Reviewer 1: Incorporating side information is a practical and valuable extension to online matrix completion. The results specializing to specific structures are helpful, primarily coming into play by bounding the quasi-dimension of the matrix relative to the side information. Presentation: Based on my own reading of the paper, I partially disagree with the reviewers about the clarity of the presentation. I think the introduction and discussion of related work are actually quite good. The writing in the main body of the paper is not necessarily bad, but I do agree that it is mathematically quite terse, which is likely to reduce the audience to experts in this area. Further comments to the authors: The authors are strongly advised to see if they can make the writing less terse. For instance, try to reduce the reliance on technical notation at least in the main paper, and elaborate more on mathematical statements in their discussion. Appendix B.2 does contain an appropriate review of the related literature, but the reviewers are correct that it does not explain which parts of the proof are different.